# Responses of unicellular predators to cope with the phototoxicity of photosynthetic prey

Akihiro Uzuka[1,2,6], Yusuke Kobayashi[1,5,6], Ryo Onuma[1,6], Shunsuke Hirooka[1], Yu Kanesaki [3], Hirofumi Yoshikawa[4], Takayuki Fujiwara[1,2] & Shin-ya Miyagishima [1,2]*

Feeding on unicellular photosynthetic organisms by unicellular eukaryotes is the base of the aquatic food chain and evolutionarily led to the establishment of photosynthetic endo-symbionts/organelles. Photosynthesis generates reactive oxygen species and damages cells; thus, photosynthetic organisms possess several mechanisms to cope with the stress. Here, we demonstrate that photosynthetic prey also exposes unicellular amoebozoan and exca-vates predators to photosynthetic oxidative stress. Upon illumination, there is a commonality in transcriptomic changes among evolutionarily distant organisms feeding on photosynthetic prey. One of the genes commonly upregulated is a horizontally transferred homolog of algal and plant genes for chlorophyll degradation/detoxification. In addition, the predators reduce their phagocytic uptake while accelerating digestion of photosynthetic prey upon illumination, reducing the number of photosynthetic cells inside the predator cells, as this also occurs in facultative endosymbiotic associations upon certain stresses. Thus, some mechanisms in predators observed here probably have been necessary for evolution of endosymbiotic associations.

---

[1] Department of Gene Function and Phenomics, National Institute of Genetics, 1111 Yata, Mishima, Shizuoka 411-8540, Japan. [2] Department of Genetics, The Graduate University for Advanced Studies, SOKENDAI, 1111 Yata, Mishima, Shizuoka 411-8540, Japan. [3] Research Institute of Green Science and Technology, Shizuoka University, 836 Ohya, Suruga, Shizuoka 422-8529, Japan. [4] Department of Bioscience, Tokyo University of Agriculture, 1-1-1 Sakuragaoka, Setagaya, Tokyo 156-8502, Japan. [5] Present address: College of Science, Ibaraki University, 2-1-1 Bunkyo, Mito, Ibaraki 310-8512, Japan. [6] These authors contributed equally: Akihiro Uzuka, Yusuke Kobayashi, Ryo Onuma.   *email: smiyagis@nig.ac.jp

Oxygen-producing photosynthesis, the primary route through which energy enters ecosystems, developed in cyanobacteria ~3 billion years ago and was then introduced into eukaryotes >1 billion years ago[1] by the establishment of plastids through cyanobacterial endosymbiosis[2–4]. This cyanobacterial endosymbiosis (primary endosymbiosis) gave rise to plastids in Archaeplastida, which consists of glaucophytes, red algae, green algae, and plants. After green and red algae had become established, plastids were further spread into several eukaryotic lineages by secondary (and tertiary) endosymbiotic events of eukaryotic algae into previously non-photosynthetic eukaryotes[2–4].

Besides the relationship between eukaryotic host cells and plastids as obligate endosymbiosis, there are several types of transient and facultative association between non-photosynthetic eukaryotic hosts and unicellular algal endosymbionts or plastids (termed kleptoplasts) derived from algae[5,6]. For example, the green ameba (*Mayorella viridis*; Amoebozoa) and several species of ciliates (Alveolata) accommodate facultative green algal endosymbionts[6]. Some species of dinoflagellates ingest unicellular eukaryotic algae and use them or their plastids as temporary plastids (kleptoplasts) for days to weeks before digesting and replacing them[6]. In certain cases such as in some dinoflagellates, the kleptoplasty is more than the simple retention of the plastids of ingested algae, but instead involves the enlargement of kleptoplasts with the aid of the nuclei of ingested algae[7]. Dinoflagellates also contain non-photosynthetic predatory species and photosynthetic species that possess plastids of red algal or other eukaryotic algal endosymbiotic origin[5]. In addition to dinoflagellates, in many other eukaryotic lineages, non-photosynthetic predatory and kleptoplastic species have been found, which are evolutionarily related to photosynthetic species that possess plastids of secondary endosymbiotic origin[5]. Based on these observations, it has been suggested that plastids were established through the phagotrophic ingestion of photosynthetic prey, transient and facultative retention, and ultimately obligate retention of photosynthetic endosymbionts by unicellular eukaryotes[2,5].

However, it appears that the transition from phagotrophy to photoautotrophy occurred through a mixture of the above-mentioned mechanisms because mixotrophic predators, which possess photosynthetic plastids of secondary endosymbiotic origin and ingest and digest prey by phagocytosis, are known in several groups of eukaryotes[6,8]. In addition, a few studies demonstrated that the early diverging prasinophyte green algae ingest and digest bacterial cells[9,10] and a very recent study suggested that phagotrophic predators, which probably possess plastids (having secondarily lost their photosynthetic ability), are a sister group of red algae[11]. These results and a comparative genomic study[12] have raised the possibility that photosynthetic eukaryotes had the capacity to perform phagocytosis when they established primary or secondary plastids, which has persisted until today, at least in some descendants[10–12]. In addition, the results of these studies suggest that the absence of phagotrophy in many photosynthetic eukaryotes is due to multiple convergent losses rather than an already established ancestral state of eukaryotes that acquired plastids[11,12].

Photosynthesis converts light energy into chemical potential and supports the life of photosynthetic organisms and other organisms through the food chain. However, electron transfer to oxygen from the photosystems ($O_2^-$) or the excitation of oxygen by chlorophyll ($^1O_2$) produces reactive oxygen species (ROS), which damage various biomolecules[13–15]. In addition, environmental stresses, such as heat, cold, drought, and light at high intensities, increase photosynthetic oxidative stress[13–15]. Thus, algae and plants have evolved various mechanisms to reduce ROS generation, quench ROS (by having higher anti-oxidant activity than non-photosynthetic organisms), and repair damaged biomolecules[13–15]; these mechanisms were probably prerequisites for eukaryotes to establish plastids and the ability to photosynthesize[4].

Although this was not examined by previous studies to our knowledge, when unicellular colorless organisms feed on photosynthetic organisms, light reaches the photosynthetic apparatus of the ingested prey during the daytime, and ROS will be generated inside the predators' cells. In addition, unregulated photosynthetic electron flow and excitation of chlorophyll molecules detached from the photosynthetic apparatus probably occur during digestion, which in turn produce higher levels of ROS inside the predators' cells. On the basis of this assumption, we investigated whether feeding on photosynthetic organisms under illumination exposes unicellular predators to oxidative stress, and if that is the case, how the predators cope with this stress. The results should yield important insights that improve our understanding of the evolutionary establishment of photosynthetic eukaryotes and of the impacts of photosynthesis on microbial food chains.

By using a newly established co-cultivation system of unicellular predators and photosynthetic or non-photosynthetic bacterial prey, we here demonstrate that feeding on photosynthetic prey under illumination exposes the unicellular predators to photosynthetic oxidative stress. We have also examined how the transcriptome and phagocytic activity of the predators change to cope with the stress. Based on the obtained results, we discuss the significance of prey–predator relationships for the further evolution of endosymbiotic associations between non-photosynthetic eukaryotes and photosynthetic endosymbionts/plastids.

## Results and Discussion

**Unicellular predators and bacterial prey co-cultivation.** To examine the effects of photosynthetic traits of prey on predators, we prepared a co-cultivation system of unicellular predators and photosynthetic or non-photosynthetic bacterial prey. As predators, we isolated three ameboid unicellular eukaryotes that feed on both photosynthetic (the cyanobacterium *Synechococcus elongatus*) and non-photosynthetic (*Escherichia coli*) bacterial prey in a sunny and shallow marsh where unicellular algae dominated (Fig. 1a, b). Nucleotide Basic Local Alignment Search Tool (BLASTn) searches using 18S rDNA sequences indicated that the isolated unicellular organisms are most closely related to *Naegleria* spp. (Excavata, class Heterolobosea), *Acanthamoeba* spp. (Amoebozoa, subclass Longamoebia), and *Vannella* spp. (Amoebozoa, subclass Flabellinia) (Supplementary Table 1). Excavata and Amoebozoa are two distinct eukaryotic supergroups[16], so these unicellular predators (hereafter, excavate *Naegleria* sp., and amoebozoans *Acanthamoeba* sp. and *Vannella* sp.) are distantly related (Fig. 1c).

In the natural habitat where we isolated the three excavate and amoebozoan species, we observed cyanobacteria that resemble *Synechococcus* spp., but they were rare in the water samples. First, we attempted to isolate and culture a non-flagellated green alga that dominated there (Fig. 1b; it did not swim and was suitable for predators to feed on), but we failed to culture it. Thus, in this study as photosynthetic prey, we used the cyanobacterium *S. elongatus*, for which a genome assembly is available[17]. We also used a previously developed procedure to prepare pale cells, which exhibit pronounced reductions of photosynthetic pigments and photosynthetic activity, in *S. elongatus*[18,19]. The genome sequence information was important for this study because we had to remove the contaminated prey sequence from RNA-seq

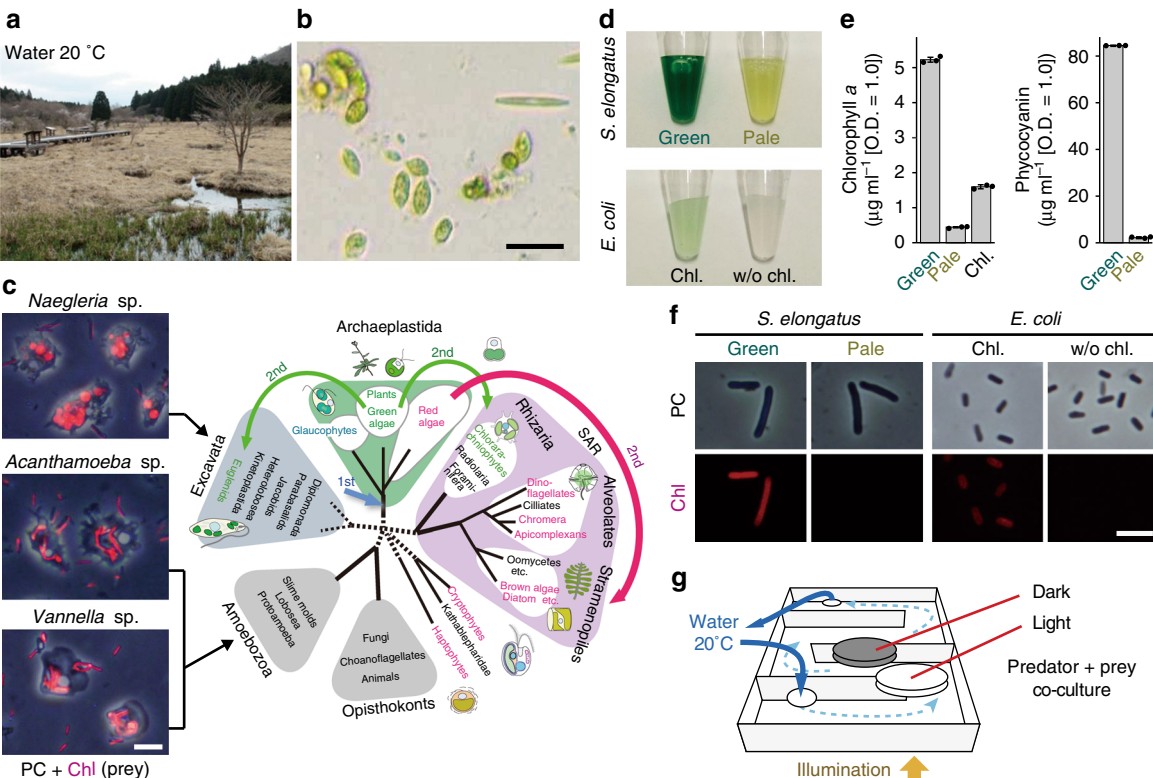

**Fig. 1 Habitat and isolation of the amoebozoan and excavate predators, preparation of bacterial prey, and co-cultivation conditions. a** Water and soil samples were collected from Kodanuki marsh in Japan to isolate ameboid predators. **b** Micrograph of the sample showing the existence of unicellular algae. Scale bar = 50 μm. **c** Three excavate and amoebozoan species that fed on the cyanobacterium *S. elongatus*. Images of phase-contrast microscopy (PC) and *S. elongatus* red chlorophyll fluorescence (Chl) are merged. The phylogenetic tree is according to ref. [16]. Scale bar = 10 μm. **d** Normal (green) and pale *S. elongatus* prey. The pale prey was prepared by cultivating the cells on nitrogen-depleted medium to reduce photosynthetic pigment levels (chlorophyll *a* and phycobilins). *E. coli* cells were incubated in acetone with (Chl) or without (w/o Chl) chlorophyll *a*, dried, and then rehydrated. **e** Chlorophyll *a* and phycocyanin levels of respective bacterial prey. The error bar represents s.d. of three independent cultures (cultured at the same time). Source data are provided as a Source Data file. **f** Micrographs of respective bacterial prey. The excitation strength for *E. coli* was 66.7 times stronger than that for *S. elongatus* for chlorophyll fluorescence (Chl) imaging. Scale bar = 5 μm. **g** The excavate and amoebozoans and bacterial prey were co-cultivated in Petri dishes. The dishes were put on a transparent box in which water at 20 °C was circulated. Dishes were illuminated from the bottom of the box.

reads to extract sequences of the excavate and amoebozoans in the following analyses. To examine the effect of the photosynthetic trait of prey on the unicellular predators, we first attempted to apply 3-(3,4-dichlorophenyl)-1,1-dimethylurea (DCMU, a widely used inhibitor of photosynthetic electron flow). However, by the addition of DCMU, the excavate and amoebozoan cells became round and subsequently died regardless of the presence or absence of cyanobacterial prey and light or dark conditions. Thus, DCMU was probably toxic to the excavate and amoebozoans. Because of this limitation, we examined the effect of the photosynthetic trait and photosynthetic pigments of prey on the unicellular predators by comparing co-culture with normal blue-green (green prey) and pale (pale prey) *S. elongatus*. Pale prey, prepared by prolonged nitrogen starvation, exhibits degradation of the bulk of its photosynthetic apparatus, including chlorophylls and phycobilins[18] (Fig. 1d, e) and has lost almost all of its photosynthetic activity (~0.1% compared with that of normal green cells)[19].

When these excavate and amoebozoan predators were cultivated in an inorganic liquid medium with bacterial prey (Fig. 1g), the concentration of bacterial prey inside the predator cells became much higher than in their surroundings (Fig. 1c: the excavate and amoebozoans that fed on green prey). All three of the excavate and amoebozoan species were used for transcriptomic analyses to obtain insights into the general responses to

photosynthetic oxidative stress. Only *Naegleria* sp. was used for other assays because, during this study, the growth of *Acanthamoeba* sp. and *Vannella* sp. became unstable after long-term storage.

**Photosynthetic prey exhibits phototoxicity to predators**. To examine whether photosynthetic prey exhibits phototoxicity to predators, *Naegleria* sp. was co-cultured with green or pale prey in an inorganic medium in the dark and then illuminated with low-(200 μE m$^{-2}$ s$^{-1}$) or high-intensity (500 μE m$^{-2}$ s$^{-1}$) light (cf. about 2000 μE m$^{-2}$ s$^{-1}$ at the surface of water in the summer). The growth rate of *Naegleria* sp. was compared with that of a co-culture kept in the dark (Fig. 2). As described later, we observed round dying cells only in the culture with the green prey under high-light conditions 60 min but not 180 min after the onset of light exposure. Based on this observation, we examined the number of *Naegleria* sp. cells 90, 180, and 360 min after the onset of light exposure (Supplementary Fig. 2) and calculated the growth rate (Fig. 2a). Because *Naegleria* sp. (also *Acanthamoeba* sp. and *Vannella* sp.) is a heterotrophic organism, it did not grow in inorganic medium without bacterial prey and gradually formed cysts after the removal of prey, as previously described[20]. In the time range of the measurements (0–360 min; Fig. 2a), the increase of the number of green *S. elongatus* cells in the low-light (×1.10 ± 0.02) and

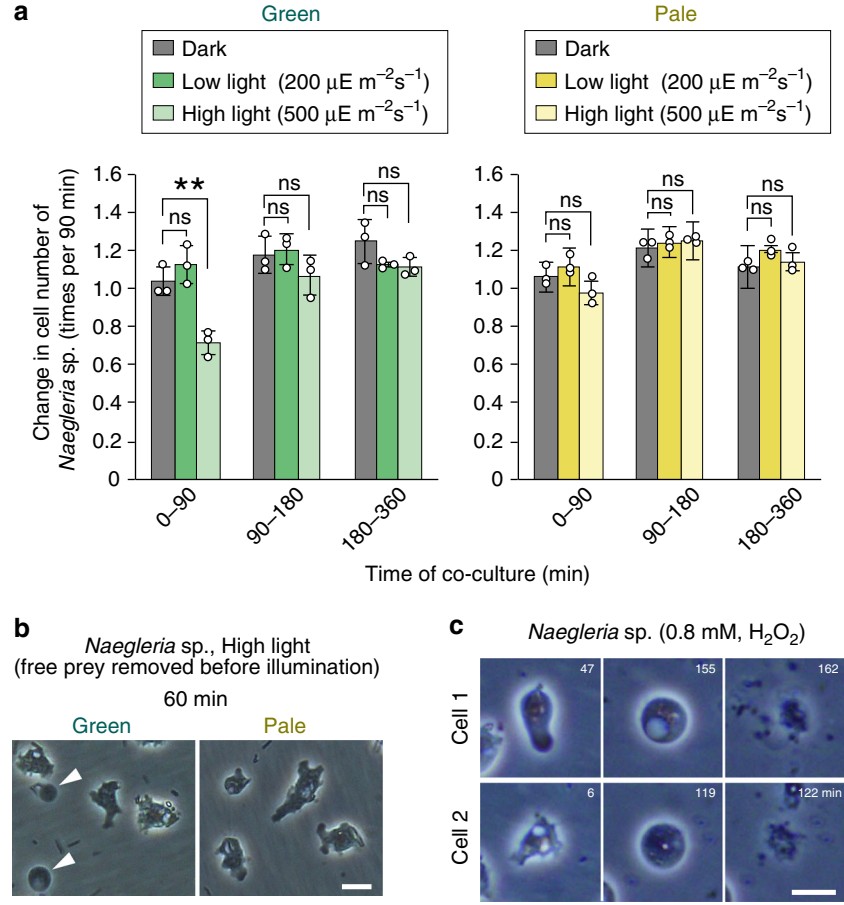

**Fig. 2 Effect of the photosynthetic trait of bacterial prey and light on *Naegleria* sp. growth.** *Naegleria* sp. feeding on green or pale *S. elongatus* under dark conditions was either cultured in the dark or transferred to low- (200 μE m$^{-2}$ s$^{-1}$) or high-light conditions (500 μE m$^{-2}$ s$^{-1}$). **a** Changes in growth rate of *Naegleria* sp. cells (fold change of the cell number per 90 min) in respective conditions. The error bar represents s.d. of three independent cultures (cultured at the same time). *$p < 0.05$; **$p < 0.005$; ns, not statistically significant (*t*-test). Changes in number of *Naegleria* sp. cells in respective conditions, from which the growth rate was calculated, are shown in Supplementary Fig. 2. Source data are provided as a Source Data file. **b** Microscopic images of *Naegleria* sp. cells 60 min after transfer to high-light conditions. Just before illumination, free bacterial prey was removed. Arrowheads indicate round dying cells. **c** Microscopic images of *Naegleria* sp. cells after treatment with 0.8 mM H$_2$O$_2$ under dark conditions. Two independent cells are shown. Scale bars = 10 μm.

high-light (× 1.11 ± 0.02) conditions was relatively slow and did not differ markedly from that in the dark (× 0.99 ± 0.04) in the inorganic medium with limited concentrations of nutrients (Supplementary Fig. 1). Likewise, the number of pale *S. elongatus* changed little in the dark (×1.00 ± 0.03), in the low-light (×1.02 ± 0.06), and high-light (× 0.99 ± 0.04) conditions (Supplementary Fig. 1). Thus, in this assay, the effect of *S. elongatus* growth and its difference depending on the culture conditions and between the green and pale cells on *Naegleria* sp. growth could be ignored.

Under dark or low-light conditions, *Naegleria* sp. proliferated in a similar manner regardless of the photosynthetic trait of the prey (green or pale) (Fig. 2a and Supplementary Fig. 2). By contrast, under high-light conditions, the number of *Naegleria* sp. cells feeding on green prey, but not on pale prey, decreased to ~70% 90 min after the onset of illumination and, after 90 min, survivors proliferated at a rate that was not significantly different from that under dark or low-light conditions (Fig. 2a and Supplementary Fig. 2). These results suggest that feeding on photosynthetic prey during the day is harmful to *Naegleria* sp. and that *Naegleria* sp. somehow adapts to the phototoxicity of photosynthetic prey in 90 min after the onset of illumination.

When the free green prey (prey that had not been ingested by *Naegleria* sp.) was removed from the co-culture before

illumination, some *Naegleria* sp. cells that had ingested the green but not the pale prey exhibited a round shape 60 min after transfer to high-light conditions (Fig. 2b). These round cells were also observed when *Naegleria* sp. was treated with exogenous H$_2$O$_2$ and then cells burst (Fig. 2c). These observations and the reduction in the number of cells in the culture with green prey under high-light conditions (Fig. 2a) suggest that the round cells observed only in the culture with green prey upon high-light exposure after the removal of free prey were dying cells and that the cell death was caused by the green prey or their digested materials inside the *Naegleria* sp. cells in the light.

**Common transcriptomic changes in predators.** To examine how the excavate and amoebozoans cope with the phototoxicity of photosynthetic prey, transcriptomic changes in the three species feeding on green prey upon illumination (200 μE m$^{-2}$ s$^{-1}$, under which most *Naegleria* sp. cells survived as shown in Fig. 2a) were examined by RNA-seq analysis (Fig. 3a, Supplementary Figs. 3–6 and Supplementary Data 1). The transcriptome 1 h after the onset of illumination was compared with that immediately before the illumination because, as discussed above, the analysis on *Naegleria* sp. growth (Fig. 2) suggested that the cells had

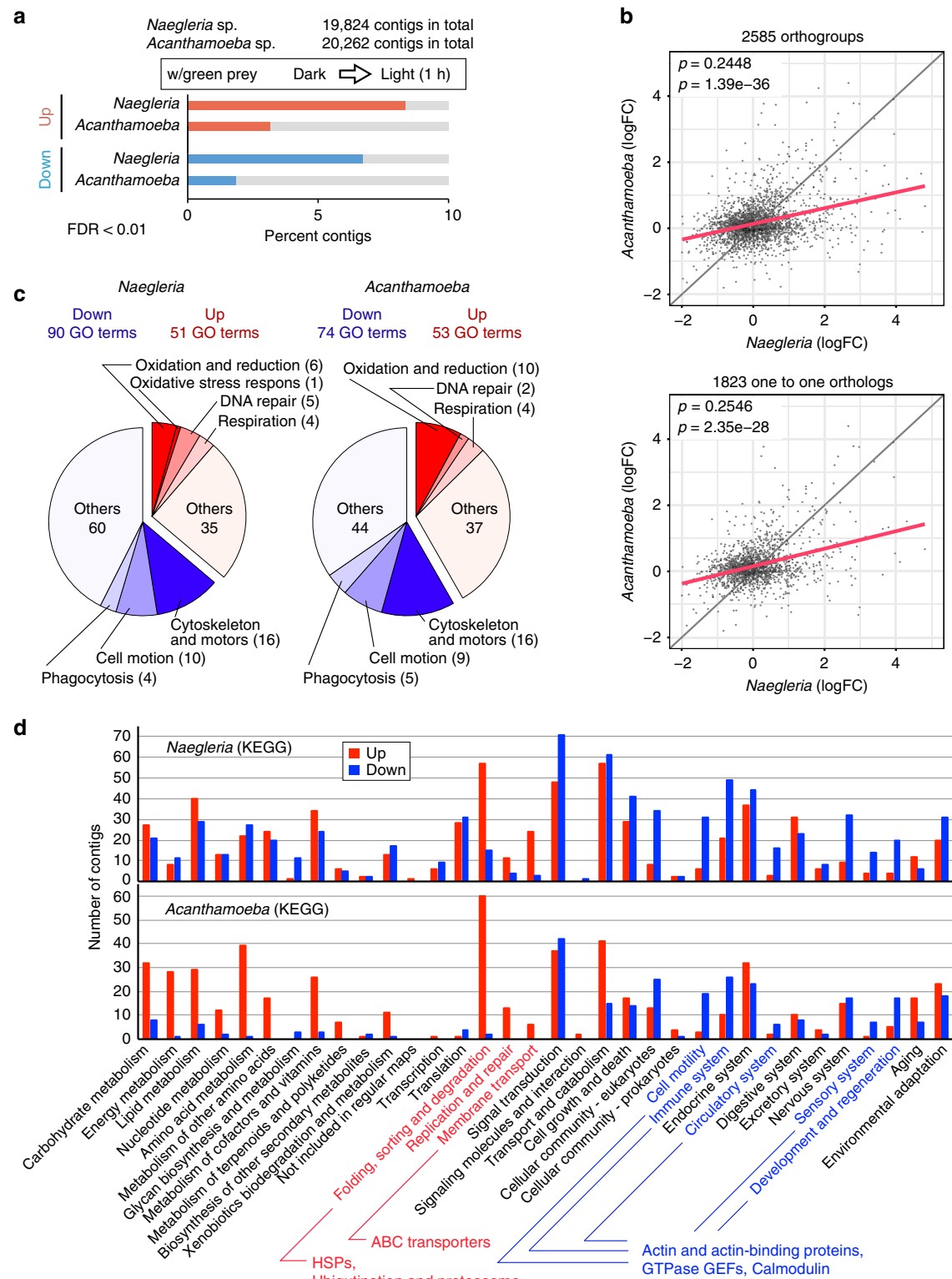

acclimated to the phototoxicity of green prey by 90 min after the onset of illumination.

Because the growth of the two species of amoebozoans became unstable after long-term storage, as described above, we obtained three independent replicates (cells were cultured independently on different days) of RNA-seq datasets (light vs. dark) in the excavate *Naegleria* sp. and the amoebozoan *Acanthamoeba* sp., but only two replicates in the amoebozoan *Vannella* sp. (Supplementary Fig. 3). Thus, statistical analyses by edgeR[21]

were applied only to datasets of *Naegleria* sp. and *Acanthamoeba* sp. The changes of mRNA levels of some genes by RNA-seq were confirmed by quantitative reverse-transcription (RT)-PCR in *Naegleria* sp. (Supplementary Fig. 6).

RNA-seq analyses [false discovery rate (FDR) < 0.01 by edgeR; three independent replicates] indicated that 8 and 7% of contigs (genes) were up- and downregulated upon illumination in the excavate *Naegleria* sp. feeding on green prey (Fig. 3a). In *Acanthamoeba* sp. feeding on green prey, 3 and 2% of contigs

**Fig. 3 Transcriptomic effect of light exposure on the excavate and amoebozoans feeding on green prey. a** Percentage of up- (red bars)/downregulated (blue bars) contigs (genes) when the excavate *Naegleria* sp. and the amoebozoan *Acanthamoeba* sp. with green prey under dark conditions were illuminated (200 µE m$^{-2}$ s$^{-1}$) for 1 h (FDR < 0.01 by edgeR; three biological replicates cultured independently on different days; Supplementary Fig. 3). **b** Scattered plot comparison of expressional changes upon illumination of orthogroups between *Naegleria* sp. and *Acanthamoeba* sp. Orthogroups shared by the two species were identified by OrthoFinder[22]. Means of log-fold changes (light vs. dark; logFC calculated with edgeR; three biological replicates) of contigs grouped in respective orthogroups (2585 in total) are plotted (top). Plots of only one-to-one orthologs (orthogroups composed of single copy genes of the two species; 1823 in total) are also shown (bottom). Spearman rank correlation coefficients ($\rho$), regression lines (red), and *p*-values (*t*-test) are shown in the graphs. **c** Content of GO terms enriched in up-/downregulated contigs (FDR < 0.01 by edgeR; three biological replicates) upon illumination in *Naegleria* sp. and *Acanthamoeba* sp. with green prey ($P < 0.05$; GOseq). Some GO terms were categorized into seven categories: cell motion, cytoskeleton and motors, DNA repair, oxidation and reduction, oxidative stress responses, phagocytosis, and respiration. The list of GO terms enriched in up-/downregulated contigs is shown in Supplementary Tables 3–6. **d** Comparison of the number of up-/downregulated contigs (FDR < 0.01 by edgeR; three biological replicates) upon illumination in *Naegleria* sp. and *Acanthamoeba* sp. with green prey whose functions were assigned to respective KEGG functional categories. The functional categories shown in red or blue indicate that more than double the number of up-/downregulated contigs were assigned to the category than down-/upregulated contigs and more than five contigs were assigned to the category in both *Naegleria* sp. and *Acanthamoeba* sp. The list of contigs assigned into respective KEGG categories is shown in Supplementary Data 2.

were up- and downregulated upon illumination (Fig. 3a). The quantitative RT-PCR analyses for 12 contigs in *Naegleria* sp. reproduced the results of RNA-seq analysis (i.e., upregulation or downregulation upon illumination) (Supplementary Figs. 4 and 6).

First, we compared whole transcriptomic changes upon illumination between *Naegleria* sp. and *Acanthamoeba* sp. To this end, each homologous gene set shared by the two species was identified as an orthogroup by OrthoFinder[22]. Then, mean of log2-fold change of each orthogroup of respective species was plotted diagonally (Fig. 3b). As a result, both the comparison of the whole orthogroups (2585 groups) and that of only one-to-one orthologs (i.e., one orthogourp was composed of single copy genes rather than multiple homologs of respective species; 1823 orthologs) exhibited a weak correlation (Spearman rank correlation coefficient; $\rho = 0.24$ and 0.25, respectively) between *Naegleria* sp. and *Acanthamoeba* sp. (Fig. 3b). The result suggests that a certain degree of transcriptomic change was shared between the two species.

To assess what kinds of functions were up-/downregulated upon illumination in these two distantly related predators feeding on green prey, gene ontology (GO) term enrichment analyses and functional classification using the Kyoto Encyclopedia of Genes and Genomes (KEGG) database[23] were performed (Fig. 3c, d and Supplementary Tables 3–6). We also checked possible functions of the up-/downregulated genes (for all three species of predators; Supplementary Fig. 4). The results indicated that GO terms that are related to oxidation and reduction, oxidative stress responses, and DNA repair were enriched exclusively in upregulated contigs in both *Naegleria* sp. and *Acanthamoeba* sp. (Fig. 3c and Supplementary Tables 3–6). In addition, in the KEGG classification, the 'replication and repair' category predominantly contained upregulated contigs in both of the species (Fig. 3d and Supplementary Data 2). These results are in agreement with the phototoxicity of the green prey as described above (Fig. 2). In addition, GO terms related to mitochondrial respiration (Fig. 3c and Supplementary Tables 3–6) in *Naegleria* sp. and *Acanthamoeba* sp. and genes encoding several monooxygenases (Supplementary Fig. 4) in *Naegleria* sp., *Acanthamoeba* sp., and *Vannella* sp. were upregulated upon illumination, presumably resulting in the consumption of oxygen that is generated by the photosystems of prey and is a source of ROS. Genes that are related to the metabolism of carotenoids, which are physical and chemical quenchers of singlet oxygen[24], were only found among the upregulated genes in *Naegleria* sp., *Acanthamoeba* sp., and *Vannella* sp. (Supplementary Fig. 4). In the KEGG classification, 'folding, sorting, and degradation' category predominantly

contained upregulated contigs in *Naegleria* sp. and *Acanthamoeba* sp. (Fig. 3d and Supplementary Data 2). This category mainly consisted of genes encoding heat-shock proteins and genes related to ubiquitination and proteasome. Also, 'membrane transport' category, which mainly consisted of components of ABC transporters, predominantly contained upregulated contigs in *Naegleria* sp. and *Acanthamoeba* sp. (Fig. 3d and Supplementary Data 2). However, the biological significance of upregulation of these categories is currently unclear.

By contrast, GO terms that are related to the cytoskeleton and motors, cell motion, and phagocytosis were enriched exclusively in the downregulated contigs in both *Acanthamoeba* sp. and *Naegleria* sp. (Fig. 3c and Supplementary Tables 3–6). In the KEGG classification, 'cell motility,' 'immune system', 'circulatory system', 'sensory system', and 'development and regeneration' categories predominantly contained downregulated contigs in both *Naegleria* sp. and *Acanthamoeba* sp. These categories mainly consisted of genes encoding actin, actin-binding proteins, and the RAS family of GTPases, which regulate actin cytoskeletal integrity, cell migration, endocytosis, and several other cellular activities, and the guanine nucleotide exchange factors, which activate RAS GTPases[25], and calmodulin (Fig. 3d and Supplementary Data 2). Consistent with these results, most of the genes encoding several types of myosin protein were downregulated in *Naegleria* sp., *Acanthamoeba* sp., and *Vannella* sp. (Supplementary Fig. 4). In addition, actin genes were also downregulated in *Naegleria* sp. and *Acanthamoeba* sp. (Supplementary Fig. 4).

The above results indicate that, upon illumination when feeding on photosynthetic prey, there was some transcriptomic up- or downregulation in common among these evolutionarily distant predators at the level of functional classification and, in some cases, at the gene level.

**Transcriptomic change is attributable to oxidative stresses.** The change in the mRNA level of some genes was likely caused by the light stimulus regardless of the photosynthetic trait of the prey. To discern changes due to illumination and those due to the photosynthetic trait of the prey, we also examined transcriptomic changes upon illumination in the predators feeding on pale prey (three independent cultures and RNA-seq analyses for *Naegleria* sp., but a single analysis for *Acanthamoeba* sp. and *Vannella* sp.) and those grown in an organic medium without bacterial prey (*Naegleria* sp. and *Acanthamoeba* sp., because *Vannella* sp. did not grow in this condition; three independent replicates for *Naegleria* sp., and a single analysis for *Acanthamoeba* sp.) (Supplementary Fig. 3).

In addition, to assess the extent to which the changes in the transcriptome observed upon illumination in *Naegleria* sp. feeding on green prey was related to photosynthetic oxidative stress, the results of green prey were compared with the effect of chlorophyll-stained *E. coli* prey (Fig. 1d–f) [comparison of light/dark ratio between the cells with chlorophyll-stained (Chl) and unstained (w/o Chl) *E. coli*], exogenous Rose Bengal (RB) (generates singlet oxygen upon illumination; comparison of light/dark ratio between the cells with or without RB in the organic medium), and $H_2O_2$ treatments (comparison between cells with or without $H_2O_2$ in the dark in the organic medium) (three independent analyses in *Naegleria* sp.; Supplementary Fig. 3).

To prepare chlorophyll-stained or unstained *E. coli* prey, *E. coli* cells were incubated in acetone with or without chlorophyll *a*, dried, and then rehydrated (Fig. 1d, e). The chlorophyll-stained *E. coli* emitted red fluorescence with blue excitation light (Fig. 1f). This observation indicates that at least some of the chlorophyll *a* molecules on the *E. coli* cells were transformed to an $S_1$ state by excitation light, which can transform to $T_1$ and produce $^1O_2/O_2^-$ by releasing energy/electrons to an oxygen molecule, leading to relaxation to $S_0$[26]. In addition, it is possible that chlorophyll *a*, which is released from *E. coli* in predator cells during digestion, and its derivatives also became toxic to predator cells[26].

The transcriptomic changes in predators upon illumination or $H_2O_2$ addition in the dark in the cultures described above were examined by RNA-seq analyses (Supplementary Fig. 5; Supplementary Data 2). In the case of *Naegleria* sp., the results of 12 contigs were validated by quantitative RT-PCR (Supplementary Fig. 6). As a result, most of the RNA-seq analyses were reproduced by quantitative RT-PCR as below. Of the 72 cases of mRNA level comparisons (six comparisons for 12 genes), 70 cases reproduced the results by RNA-seq analyses (i.e., either upregulation, downregulation or unchanged of respective mRNA levels) (Supplementary Figs. 5, 6). The two cases of exceptions were the results of two glutathione peroxidase genes upon $H_2O_2$ treatment, in which quantitative RT-PCR showed downregulation (Supplementary Fig. 6) while the changes were not statistically significant in RNA-seq analyses (Supplementary Fig. 5). This is likely due to the large deviation of results of $H_2O_2$ treatment among the independent experimental replicates as described below.

By RNA-seq analyses, the majority (74 and 58%) of mRNAs that were up-/downregulated upon illumination in *Naegleria* sp. feeding on green prey (Fig. 3) were also up-/downregulated (FDR < 0.01, edgeR; three independent replicates) by a treatment with either of chlorophyll *a* (log2-fold change of light/dark in 'Chl' was larger than that of light/dark in 'w/o Chl'), RB (log2-fold change of light/dark in culture with RB was larger than that of light/dark in culture without RB), or $H_2O_2$ (Fig. 4a). These results suggest that the majority of the changes in mRNA levels in *Naegleria* sp. feeding on green prey upon illumination are related to oxidative stress and/or effects of chlorophyll *a*.

To further compare the effect of light stimulus and oxidative stress on the transcriptomic change in *Naegleria* sp. feeding on green prey, the transcriptomic changes upon illumination (or addition of $H_2O_2$ in the dark) in respective culture conditions were compared in a two dimensional map by the t-distributed stochastic neighbor embedding (t-SNE)[27] (Fig. 4b). The changes upon illumination with reduced (with pale prey) and no (without prey, with unstained *E. coli*) photosynthetic traits of prey were similar (Fig. 4b). In contrast, the transcriptomic change with green prey was placed at a different position (Fig. 4b), suggesting that the transcriptomic change when feeding on green prey is not solely attributable to the light stimulus. In addition, the change with chlorophyll-stained *E. coli* was positioned close to the change with green prey (Fig. 4b). Although the changes with RB or upon $H_2O_2$ addition largely deviated among three sets of replicates, they were positioned between the change with green prey and those with reduced or no photosynthetic traits of prey (Fig. 4b). This observation is apparently consistent with the Venn diagram in which the transcriptomic change with green prey was shared more by the chlorophyll treatment than RB or $H_2O_2$ treatments (Fig. 4a).

Although the transcriptomic change upon illumination in *Naegleria* sp. feeding on green prey was similar to that due to ROS and/or chlorophyll *a* treatments (Fig. 4a, b), the response modes differed among the genes. For example, mRNAs encoding glutathione peroxidase, which is known to reduce oxidative damage[28], and *COX15* mRNA, which is involved in heme A synthesis[29] and in cytochrome c oxidase assembly in the mitochondrial respiratory chain[30], were upregulated in the three excavate and amoebozoan species feeding on green prey upon illumination (Fig. 4c and Supplementary Fig. 4). In *Naegleria* sp., the genes were also upregulated upon illumination feeding on pale prey and when they were cultured without prey (Fig. 4c and Supplementary Fig. 6). Thus, the genes could be upregulated by the light stimulus regardless of the photosynthetic trait of the prey. However, mRNAs encoding some copies of glutathione peroxidase (Fig. 4c and Supplementary Fig. 6), but not *COX15* (Supplementary Fig. 6) mRNA, were also upregulated by $H_2O_2$ treatment under dark conditions. Thus, the genes can be upregulated by either light stimulus or oxidative stress.

The mRNAs of phytoene desaturase, which is involved in carotenoid synthesis[31], were upregulated in *Naegleria* sp. and *Acanthamoeba* sp. feeding on green prey upon illumination (Fig. 4c and Supplementary Fig. 4). In *Naegleria* sp., the gene was upregulated upon illumination regardless of the photosynthetic trait of the prey (i.e., feeding on pale prey and/or culture without prey) (Fig. 4c). However, the gene was also upregulated by $H_2O_2$ treatment under dark conditions (Fig. 4c). In addition, the gene was more highly upregulated upon illumination in the culture with RB than that without this chemical and also more highly upregulated in cells feeding on chlorophyll-stained *E. coli* than those feeding on unstained *E. coli* (Fig. 4c). Thus, the gene can be upregulated by either light stimulus or oxidative stress/chlorophyll *a*.

Type I and II myosin genes were downregulated upon illumination with feeding on green prey, but not with feeding on pale prey or culture without prey (Fig. 4c and Supplementary Fig. 6). In addition, the genes were also downregulated in *Naegleria* sp. upon feeding on *E. coli* stained with chlorophyll *a*, but not unstained *E. coli* (upon illumination), or when treated with RB (upon illumination) or $H_2O_2$ (in the dark) (Fig. 4c and Supplementary Fig. 6). Thus, these genes were downregulated specifically by oxidative stresses or chlorophyll *a* treatment.

**Regulation of number of photosynthetic prey inside cells.** Genes encoding actin and myosin proteins, which play roles in phagocytosis beneath the cell membrane[32], and genes related to phagocytosis were downregulated upon illumination in the three excavate and amoebozoan species feeding on green prey and with ROS treatments in the excavate *Naegleria* sp. (Figs. 3c and 4c and Supplementary Figs. 4–6). On the basis of these results, we examined the effect of the photosynthetic trait of prey on phagocytic activity in *Naegleria* sp. To this end, *Naegleria* sp. was pre-cultured with green or pale prey under dark conditions and then further incubated under dark or light conditions (200 $\mu E\,m^{-2}\,s^{-1}$) for 1 h. Next, fluorescent latex beads of 1 $\mu m$ in diameter were added to the culture to quantify phagocytic activity. In this assay, *Naegleria* sp. cells were able to ingest up to four beads in 1 h (Fig. 5a, b). When

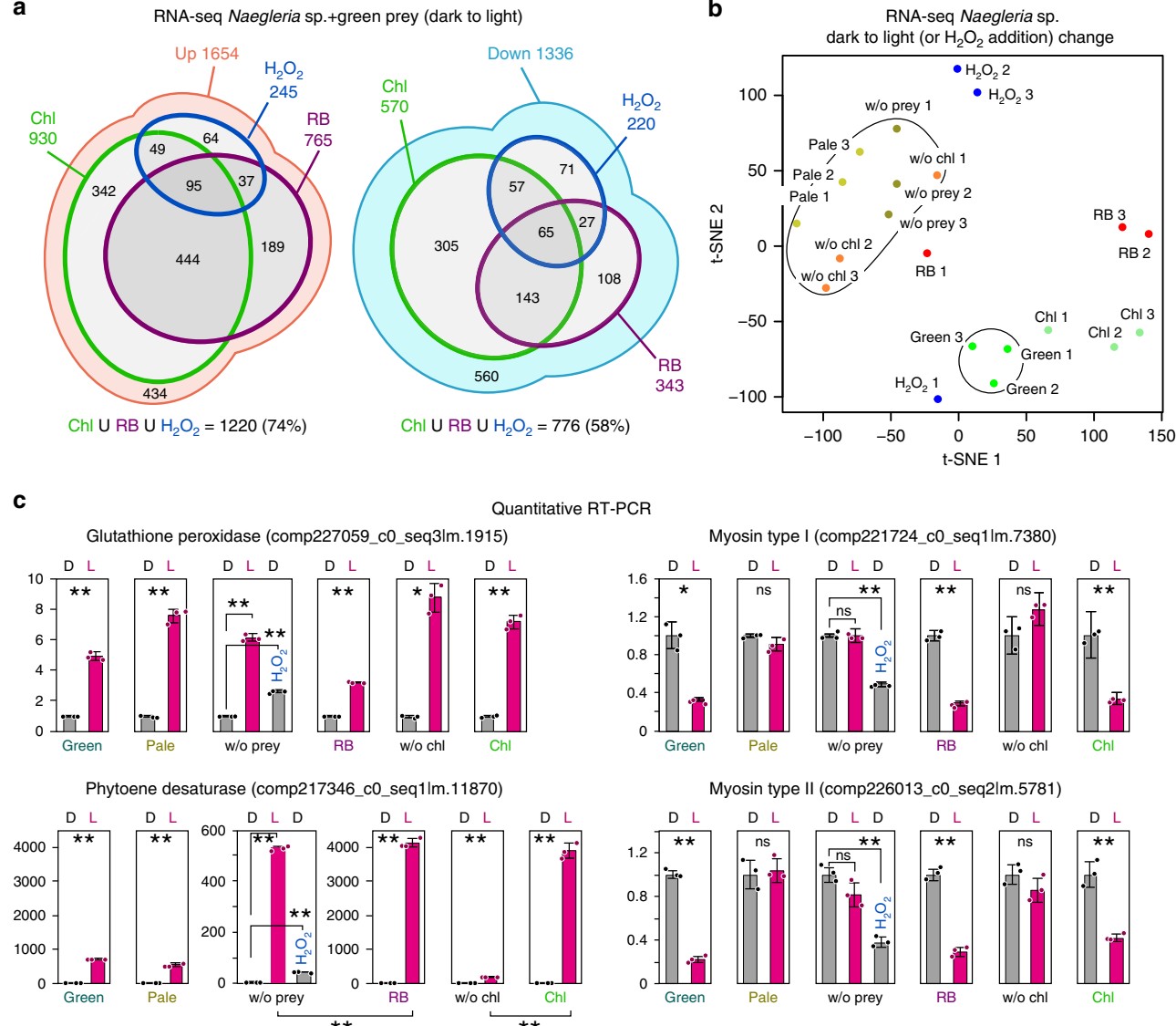

**Fig. 4 Transcriptomic effect of the photosynthetic trait of bacterial prey, chlorophyll, and ROS on *Naegleria* sp. a** The content of up-/downregulated contigs upon illumination in *Naegleria* sp. with green prey that were also up-/downregulated (FDR < 0.01; edgeR) in *Naegleria* sp. by ingestion of chlorophyll *a* (log2-fold change of light/dark in 'Chl' based on normalized count data by TCC package[69], which used edgeR, was larger than that of light/dark in 'w/o Chl'; *p* < 0.05, *t*-test; three biological replicates cultured on different days), by 30 nM Rose Bengal (RB) treatment (producing $^1O_2$ in the light) [log2-fold change of light/dark in culture with RB was larger than that of light/dark in culture without RB (without prey); *p* < 0.05, *t*-test; three biological replicates cultured on different days], or by a treatment with 0.1 mM $H_2O_2$ (FDR < 0.01, compared with a culture without $H_2O_2$; three biological replicates cultured on different days). **b** t-SNE two dimensional representation of transcriptomic changes upon illumination (or $H_2O_2$ addition in the dark) in respective culture conditions. Log2-fold changes (light vs. dark or with vs. without $H_2O_2$) of contigs in respective culture conditions are represented. **c** Quantitative RT-PCR analyses showing examples of changes in mRNA levels of contigs. *EF1-alpha* was used as an internal control. The level under dark conditions (or before $H_2O_2$ addition) was defined as 1.0. The error bar represents s.d. of three replicates (three sets of RNA samples prepared from three sets of *Naegleria* sp. cultures that were performed at the same time). **p* < 0.05; ***p* < 0.005; ns, not statistically significant (*t*-test). Results of other genes are shown in Supplementary Fig. 6. Examples of changes in levels (FPKM values by RNA-seq analyses) of mRNA in respective culture conditions in *Naegleria* sp. are shown in Supplementary Fig. 5. The details of the culture conditions are described in Supplementary Fig. 3. Source data are provided as a Source Data file.

*Naegleria* sp. fed on pale prey, the number of beads that were ingested by *Naegleria* sp. cells was similar between light and dark conditions (Fig. 5b). By contrast, when *Naegleria* sp. fed on green prey, the number of beads that were ingested by *Naegleria* sp. cells under light conditions was significantly lower than that under dark conditions (Fig. 5b). These results suggest that phagocytic activity decreased when *Naegleria* sp. fed on photosynthetic prey under illumination.

The results then raised the question of how *Naegleria* sp. copes with photosynthetic prey that has been ingested during a dark period but has not been digested when illuminated. To answer this question, the digestion rate of ingested prey was compared in *Naegleria* sp. feeding on green or pale prey under light or dark conditions. To quantify such digestion, cell membranes of intact green and pale prey were stained with the fluorescent dye FM1–43. FM1–43, which is water-soluble and nontoxic to cells, is

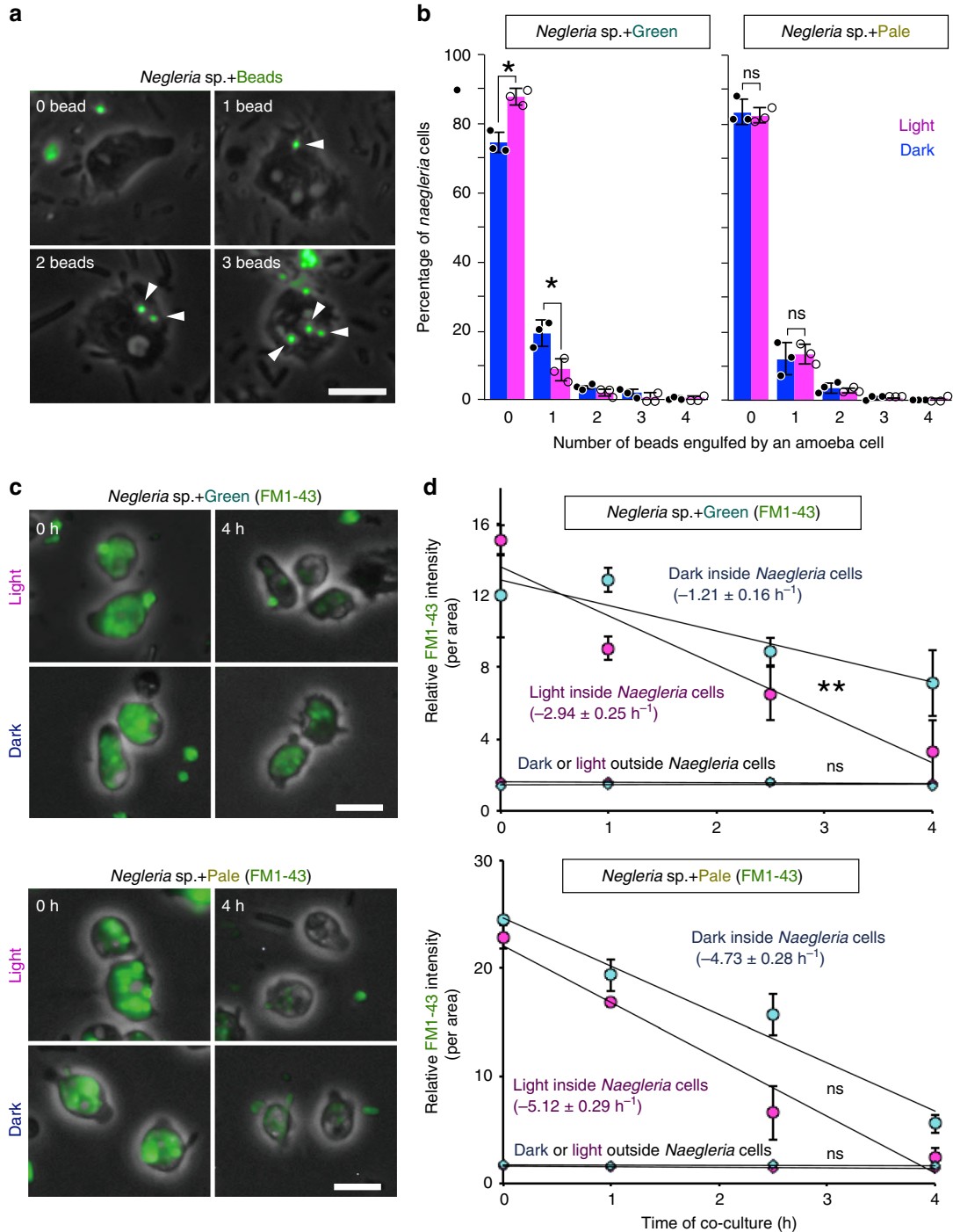

**Fig. 5 Effect of the photosynthetic trait of bacterial prey on ingestion and digestion of prey by *Naegleria* sp. a, b** *Naegleria* sp. was co-cultured with green or pale *S. elongatus* prey under dark conditions and then either kept in the dark or illuminated (200 µE m$^{-2}$ s$^{-1}$; light) for 1 h. After that, the culture was incubated under dark or light conditions with green fluorescent beads (1 µm in diameter) for 1 h. **a** Micrographs of *Naegleria* sp. cells that ingested 0–3 beads (green). Scale bar = 10 µm. **b** Number of fluorescent beads ingested by *Naegleria* sp. cells in respective co-cultures. The error bar represents s.d. of three independent cultures (cultured at the same time). *$p < 0.05$; ns, not statistically significant (*t*-test). Source data are provided as a Source Data file. **c, d** *Naegleria* sp. was co-cultured with green or pale prey (cell membrane was stained with the green fluorescent FM1–43) in the dark and then immobilized on MAS-coated glass so as not to newly ingest bacterial prey. Then (hour 0), the cells were kept under dark or illuminated (200 µE m$^{-2}$ s$^{-1}$; light) for 4 h. **c** *Naegleria* sp. cells (phase-contrast microscopy) digesting green or pale *S. elongatus* prey stained with green FM1–43 (green fluorescence) under dark or light conditions. Scale bar = 10 µm. **d** Decrease in FM1–43 fluorescence inside *Naegleria* sp. cells in respective conditions. Change in the fluorescence of free prey outside *Naegleria* sp. cells was also determined as a control. The error bar represents s.d. of three independent cultures (cultured at the same time). *$p < 0.05$; **$p < 0.005$; ns, not statistically significant (*t*-test; inclination in the light vs. that in the dark). Source data are provided as a Source Data file.

not fluorescent in aqueous phase but becomes fluorescent upon its insertion into the cell membrane[33]. The fluorescence of FM1–43-stained green and pale prey was stable for at least a week in our experimental conditions. Thus, the reduction of fluorescence after the prey was ingested by *Naegleria* sp. cells probably reflects digestion of the cell membrane of the prey by *Naegleria* sp.

*Naegleria* sp. cells were pre-cultured under dark conditions with green or pale prey stained with FM1–43 for the cells to ingest the prey. Then, the co-culture was immobilized on adhesive slide glass to inhibit the *Naegleria* sp. cells from continuing to ingest bacterial prey and further incubated under dark or low-light conditions ($200\,\mu E\,m^{-2}\,s^{-1}$) for 4 h (Fig. 5c, d). In our experimental conditions, the fluorescent intensity of the free prey outside *Naegleria* sp. cells was kept almost constant both in the dark and light in the time range of the measurements (Fig. 5d). Thus, the reduction of fluorescence was attributable to digestion of cell membrane rather than bleaching of the FM1–43 fluorescence in the cell membrane of prey.

When the digestion rate was compared between light and dark conditions, the rates were similar in *Naegleria* sp. feeding on pale prey (Fig. 5c, d). By contrast, the digestion rate was about two times higher under light than under dark conditions in *Naegleria* sp. cells feeding on green prey (Fig. 5c, d). Thus, *Naegleria* sp. cells decrease their uptake and accelerate their digestion of green prey but not pale prey upon illumination, which leads to a decrease in the number of photosynthetic prey inside predator cells and thus probably reduces oxidative stress.

**Phaeophorbide *a* oxygenase-like genes of predators**. Finally, we further manually checked contigs (genes) that were commonly up- or downregulated in the three excavate and amoebozoan species. As a result, we found genes encoding proteins that possess both Rieske [2Fe-2S] iron-sulfur domain and Phaeophorbide *a* oxygenase (PAO)-like domain. The excavate *Naegleria* sp. had two copies of this gene and *Acanthamoeba* sp. and *Vannella* sp. had a single copy of it, with all copies being upregulated upon illumination with feeding on green prey (Fig. 6).

PAO also possesses Rieske [2Fe-2S] iron-sulfur domain and is involved in chlorophyll breakdown and detoxification during plant senescence[34]. During the breakdown process, phytol and the central Mg atom of chlorophyll *a* are removed and then the ring structure of the resultant pheophorbide *a* is oxygenolytically opened by PAO[34]. The resultant red chlorophyll catabolite is further converted to primary fluorescent chlorophyll catabolite (pFCC). Finally, pFCC is modified and transported into the vacuole[34]. In addition to PAO, the genome of the land plant *Arabidopsis thaliana* encodes three other proteins that possess both Rieske [2Fe-2S] iron-sulfur domain and PAO-like domain, namely, chlorophyllide *a* oxygenase (CAO)[35,36], Translocon at the inner chloroplast envelope 55 (TIC55)[37], and Protochlorophyllide-dependent translocon component 52 (PTC52)[38,39]. CAO is involved in chlorophyll *b* formation from chlorophyll *a*[35,36,40]. Tic55 was originally identified as a component of a protein translocon at the chloroplast inner envelope membrane and was recently found to be a hydroxylase of pFCC[37]. PTC52 was identified as a plastid envelope protein that interacts with cytosolic precursor of protochlorophyllide oxidoreductase (pPOR) during the translocation of pPOR from the cytosol to the chloroplast[39]. In addition, PTC52 was shown to function as a protochlorophyllide (Pchlide) *a* oxygenase, which is probably essential for controlling Pchlide homeostasis and pPOR import[38,39]. Thus, all of the four *A. thaliana* proteins that possess both Rieske [2Fe-2S] iron-sulfur domain and PAO-like domain are involved in chlorophyll metabolism.

To obtain insights into the possible function and origin of PAO-like proteins in the excavate and amoebozoans, we conducted a phylogenetic analysis of PAO-like proteins (Fig. 6a; the *Vannella* sp. sequence was omitted from the analysis because we obtained the sequence of only its C-terminal half). Homologs of PAO were widely found in cyanobacteria and eukaryotes that possess plastids of primary and secondary endosymbiotic origins (Fig. 6a), although their functions other than in plants (described above) and CAO in some green algae[40,41] have not been examined. In addition to the excavate and amoebozoans that were isolated and used in this study and their related and completely sequenced species (*Naegleria gruberi* and *Acanthamoeba castellanii*), homologs were found in a few gamma-proteobacterial species and certain species of fungus that swim using flagella and/or exhibit ameboid morphology[42,43] (Fig. 6a).

The conservation of these PAO-like proteins in photosynthetic organisms and their distribution in a limited number of heterotrophic organisms (Fig. 6a) suggest that the genes were horizontally transferred from photosynthetic organisms to the excavate and amoebozoans. Based on the predicted amino acid sequences, a membrane-spanning domain exists at the N-terminal end in the two *Naegleria* sp. proteins while the *Acanthamoeba* sp. protein possess that at the C-terminal end (Fig. 6a). This opposite location of a membrane-spanning domain suggests that two genes in the excavate *Naegleria* sp. and one gene in the amoebozoan *Acanthamoeba* sp. have been acquired independently although the values supporting the independent origins by the phylogenetic analysis were relatively low (a bootstrap value of 78 by the maximum-likelihood method and a posterior probability of 0.99 by the Bayesian analysis) (Fig. 6a).

In the phylogenetic analysis, both the *Naegleria* sp. and *Acanthamoeba* sp. proteins were grouped with CAO of *A. thaliana* together with proteins in fungi and some homologs of photosynthetic organisms (bootstrap value of 99). However, CAO does not possess any membrane-spanning domain, unlike *Naegleria* sp. and *Acanthamoeba* sp. proteins (Fig. 6a). In addition, the identity between *Naegleria* sp./*Acanthamoeba* sp. proteins and *A. thaliana* CAO was only 25–28%. Thus, proteins in these predators likely exert certain activity that is related to the degradation/detoxification of chlorophylls or their derivatives, but different from CAO activity. Supporting this possibility, one gene in *Naegleria* sp. was also upregulated by $H_2O_2$ treatment without illumination and the magnitude of upregulation was higher in culture with chlorophyll *a* than in that without chlorophyll *a* upon illumination (Fig. 6b and Supplementary Fig. 6).

**Strategies to cope with photosynthetic oxidative stress**. The detection of ROS is often difficult because of their very short half-life and the immediate and sequential transmission of their reactivities to other molecules[44]. In fact, we compared the $H_2O_2$ concentration in the medium (with the Amplex Red Enzyme Assay; Thermo Fisher Scientific) between co-culture of *Naegleria* sp. with green prey in the dark and that in the light, but we did not detect any significant difference. We also attempted to detect ROS generation by prey inside *Naegleria* sp. cells upon illumination with Singlet Oxygen Sensor Green (Thermo Fisher Scientific). However, because of the fluorescence of Oxygen Sensor Green that increased with light alone without *Naegleria* sp. cells and prey, we could not reliably evaluate ROS generation. When CellROX Green or Orange (Thermo Fisher Scientific) was applied, mitochondrial nucleoids or mitochondria of *Naegleria* sp. feeding on green prey emitted weak fluorescence, respectively, both in the light and dark. However, we could not detect any difference in the fluorescent intensity between the two conditions.

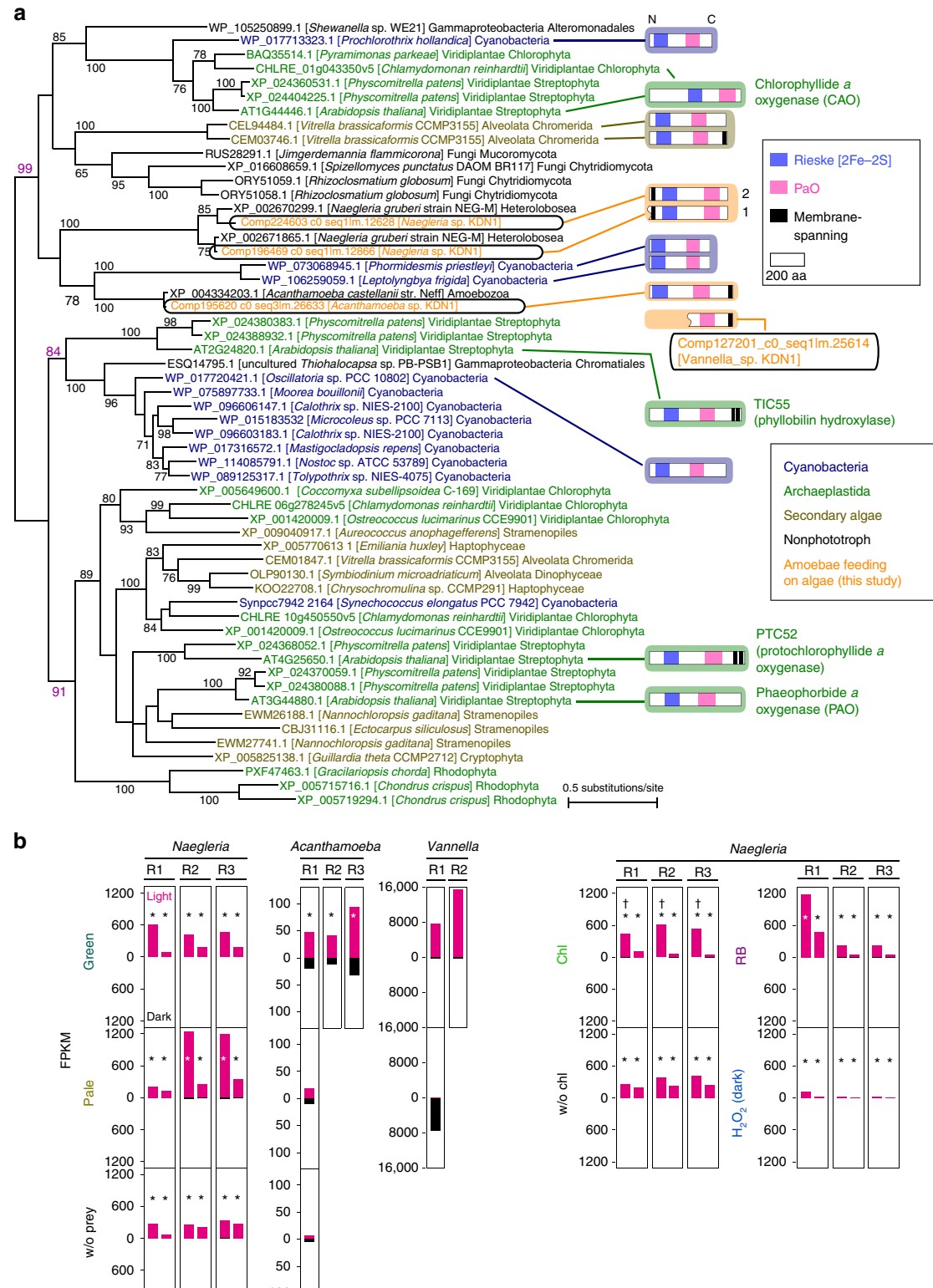

However, (i) cell death upon exposure to high light (Fig. 2), (ii) upregulation of genes related to oxidation/reduction and DNA repair in the excavate and amoebozoans feeding on green prey upon illumination (Fig. 3), and (iii) the result that the majority of transcriptomic change was shared with responses to ROS treatments (Fig. 4a) suggest that photosynthetic prey exhibits phototoxicity to unicellular colorless predators, which is most likely attributable to oxidative stresses.

The comparison of transcriptomic changes showed that the change upon illumination in the predator feeding on the photosynthetic prey was different from that without photosynthetic traits of prey (Fig. 4b) and that the change is largely shared with responses to chlorophyll/ROS treatments (Fig. 4a). However, some genes were upregulated upon illumination regardless of photosynthetic traits of prey (Fig. 4c and Supplementary Figs. 4–6) although many of these genes were also upregulated by ROS

**Fig. 6 Phylogenetic relationship of Rieske-type oxygenases that possess pheophorbide a oxygenase (PAO)-like domain and effect of the photosynthetic trait of prey, chlorophyll, and ROS on their mRNA levels in the three excavate and amoebozoan species. a** The tree was constructed by the maximum-likelihood (ML) method (RaxML 7.0.4). ML bootstrap values of >50% obtained by RaxML are shown above the branches, which were also supported by posterior probabilities of >0.95 by Bayesian analysis (MrBayes 3.2.6). The accession numbers or locus IDs of the sequences are shown along with the names of the species. The branch lengths reflect the evolutionary distances indicated by the scale bar. Predicted structures are also shown for some proteins. Putative transmembrane domain (predicted by TMHMM 2.0), Rieske [2Fe-2S] iron-sulfur domain, and PAO-like domain (predicted by Pfam 32.0) are shown. Wavy lines at the left edge indicate that the full *orf* sequence was not available and thus the N-terminal sequence was not clear. **b** Comparison of levels of mRNA (FPKM values by RNA-seq analyses) encoding PAO-like proteins in respective culture conditions in *Naegleria* sp., *Acanthamoeba* sp., and *Vannella* sp. The details of the culture conditions are described in Supplementary Fig. 3. Each bar corresponds to one contig. *, FDR < 0.01 (edgeR; only in cases in which data of three biological replicates, R1, R2, and R3, of independent cultures on different days were obtained); †, Log2-fold change (light vs. dark) in 'Chl' was higher than that in 'w/o Chl' based on normalized count data by the TCC package, which uses edgeR (*p* < 0.05, *t*-test; three biological replicates cultured on different days). Results of quantitative RT-PCR for *Naegleria* sp. are shown in Supplementary Fig. 6.

treatment without illumination. The mechanisms of the response to light stimuli without photosynthetic oxidative stress are unclear at this point. However, a plausible idea is that these changes of some genes would be programmed upon illumination to cope with 'expected' phototoxicity of prey in nature.

In this study, we used the cyanobacterium *S. elongatus* as photosynthetic prey because of the difficulty in culturing the unicellular algae that dominated the natural habitat and the availability of genomic information and a procedure for preparing pale prey in *S. elongatus*. Thus, the co-culture systems that we used do not completely mimic conditions in the wild. The changes in the mRNA levels of some genes in predators observed here are likely specific to *S. elongatus* prey and likely do not occur in the predators feeding on other unicellular cyanobacteria or eukaryotic algae in natural habitats. However, as discussed above, the majority of the transcriptomic changes upon illumination in *Naegleria* sp. feeding on green *S. elongatus* are attributed to ROS and/or ingestion of chlorophyll *a*, which should be common to photosynthetic prey containing chlorophyll *a* and photosynthetic apparatus, regardless of the lineage and species.

A certain degree of transcriptomic change was shared among the three evolutionarily distant excavate and amoebozoan predators feeding on photosynthetic prey upon illumination, in terms of the associated functional categories (GO terms and KEGG classification) and, in some cases, at the gene level (Fig. 3). However, detailed repertoires of genes that are up-/down-regulated differed among the three species (Supplementary Fig. 4). In addition, the phylogenetic analysis suggested that PAO-like genes, which are likely involved in the degradation/detoxification of chlorophylls of ingested prey, were acquired by the excavate and amoebozoans through independent horizontal gene transfer (HGT) events (Fig. 6). These results suggest that the mechanisms required for coping with the phototoxicity of photosynthetic prey evolved independently in the excavate and amoebozoans.

The commonalities of prey–predator relationships and the facultative and obligate endosymbiotic associations of eukaryotes with photosynthetic organelles/endosymbionts include that a host/predator (i) reduces light absorption by photosynthetic pigments (chlorophylls and phycobilins) in the cell (Figs. 5 and 7) in addition to (ii) coping with photosynthetic oxidative stress via redox enzymes and damage repair under stressful conditions (Figs. 3 and 7)[13–15].

Regarding the reduced uptake and rapid digestion of green prey by unicellular predators upon illumination (Fig. 5), the digestion or expelling of facultative algal endosymbionts has been observed in corals[45] and green paramecia[46] under stressful conditions such as high light (Fig. 7). Thus, reducing the amount of photosynthetic prey/endosymbinots in the cells is probably a common strategy to reduce stresses caused by prey/endosymbionts in eukaryotes feeding on/accommodating cyanobacteria or eukaryotic algae. By contrast, it is known that eukaryotic algae or

sessile land plants, which permanently possess plastids, escape from high light by swimming or gliding[47,48] or they relocate their plastids in the cells[49], respectively, in addition to reducing chlorophyll levels to minimize light absorption under high-light conditions[50,51] (Fig. 7). Thus, such a change in strategy was probably important for eukaryotes to permanently possess plastids. However, it remains unclear whether the endosymbiont expulsion or degradation is directly triggered by oxidative stress or rather by some dysfunctional metabolic interactions that result in oxidative stresses in parallel.

PAO-like proteins, which are involved in chlorophyll metabolism, are widely distributed in photosynthetic organisms including cyanobacteria and eukaryotes that possess plastids of primary and secondary endosymbiotic origin (Fig. 6). Conventionally, this type of distribution of the proteins suggests that these proteins of cyanobacterial origin spread into photosynthetic eukaryotes by endosymbiotic gene transfer (EGT) from endosymbiont to host genomes through primary and secondary endosymbiotic events. However, we have found PAO-like genes probably of HGT origin in three evolutionarily distant excavate and amoebozoans feeding on photosynthetic prey, which likely play roles in the degradation/detoxification of chlorophylls derived from prey during digestion (Fig. 6). This raises the possibility that the genes were distributed to some eukaryotic lineages before the acquisition of their photosynthetic endosymbionts/plastids through HGT from photosynthetic prey, rather than EGT from photosynthetic endosymbionts. In the phylogenetic tree, the origins of PAO-like proteins of excavates and amoebozoans are apparently different from the proteins of organisms that possess plastids of primary or secondary endosymbiotic origin (Fig. 6a). The cyanobacterial proteins and those of primary plastid lineages were divided to multiple clades and, based on the topology, there were probably several gene transfers to primary plastid lineages (Fig. 6a). Thus, even in the acquisition of PAO-like proteins, several gene transfers probably took place stepwise during the establishment of primary plastids in contrast to the excavates and amoebozoans that experienced only single gene transfers.

In summary, prototypes of some mechanisms that are required for endosymbiotic associations between non-photosynthetic eukaryotic hosts and photosynthetic endosymbionts, such as the reduction of photosynthetic endosymbionts/pigments in the cells to cope with photosynthetic oxidative stress, would have developed in their predatory ancestors.

## Methods

**Isolation and identification of unicellular predators**. Water with unicellular photosynthetic organisms was sampled from the surface of mud and water plants at sunny points in Kodanuki marsh, Fujinomiya, Shizuoka, Japan (35°21′7″N, 138° 33′22″E) in April 2013 (the water was 20 °C and approximately pH 7.0). For ameboid organisms to proliferate dominantly, *E. coli* DH-5α (this strain was used for this purpose only in this study) was added to ~20 mL of water samples as prey and incubated at 20 °C under illumination (20 μE m$^{-2}$ s$^{-1}$) in 50 mL tubes for

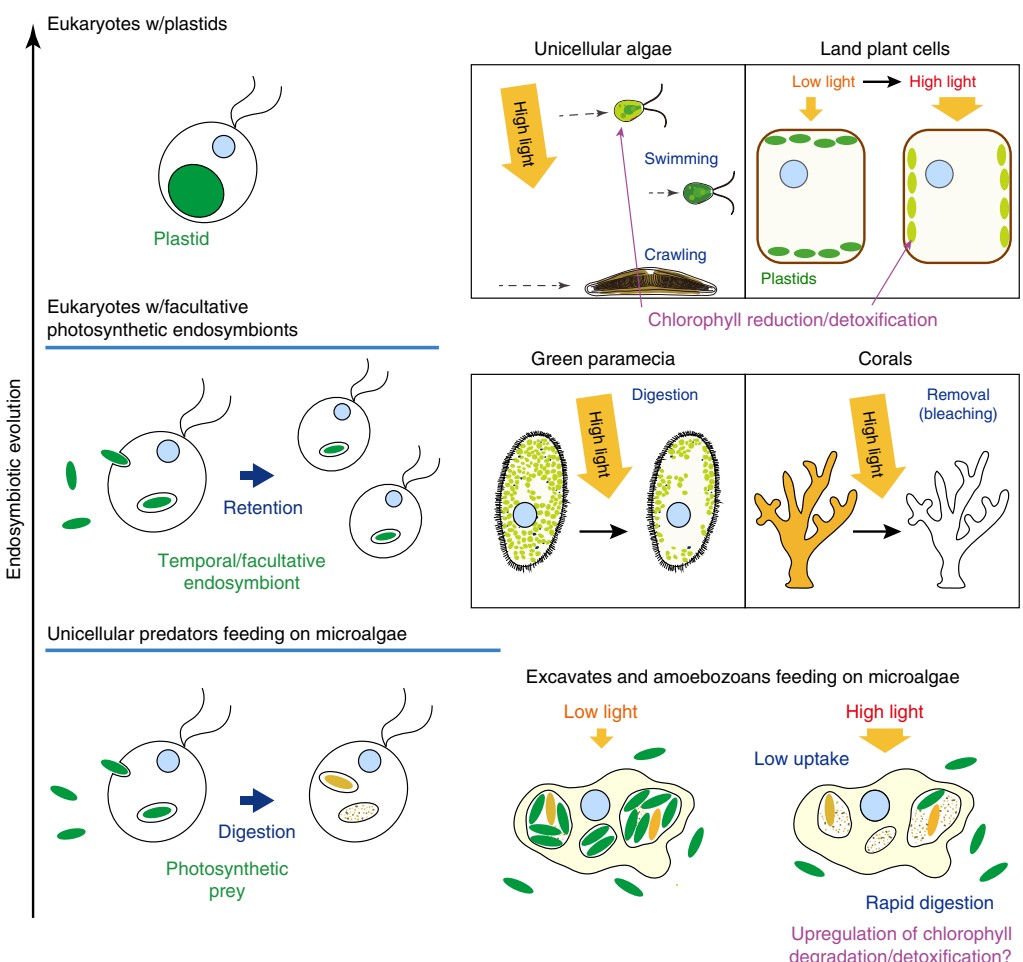

**Fig. 7 Comparison of mechanisms to reduce light absorption by photosynthetic apparatus in unicellular predators, eukaryotes accommodating facultative photosynthetic endosymbionts, algae, and plants.** When the photosynthetic apparatus absorbs excess light energy, ROS production is elevated. When eukaryotic algae are exposed to high light, they escape from the light by swimming or crawling to reduce light absorption by plastids. When land plants are exposed to high light, the leaf cell changes the positioning of plastids to reduce light absorption by plastids. A reduction in chlorophyll level also occurs in algae and plants to reduce light absorption. Regarding eukaryotes accommodating facultative photosynthetic endosymbionts, when green paramecium is exposed to oxidative stress, they digest algal endosymbionts. When corals are exposed to high light, they remove the algal endosymbionts from the cells. Regarding unicellular predators feeding on algae, when they are illuminated, they reduce their uptake of photosynthetic prey and accelerate their digestion of photosynthetic prey, thus leading to a reduction of photosynthetic prey inside the cells. In addition, PAO-like genes of HGT origin, which are likely involved in degradation/detoxification, are upregulated upon illumination in the unicellular predators feeding on algae.

6 days. The results showed that several kinds of ameboid organism proliferated in some water samples. Then, for ameboid organisms feeding on unicellular photosynthetic organisms to proliferate dominantly, the mixture of ameboid organisms was inoculated in BG-11 liquid medium, an inorganic medium widely used for the cultivation of freshwater cyanobacteria[52], and supplemented with the cyanobacterium *S. elongatus* PCC 7942 at 20 °C under illumination for 6 days. To isolate respective cells, the culture was serially diluted with BG-11 medium supplemented with *S. elongatus* in 96-well plates and incubated at 20 °C under illumination. Ameboid organisms that proliferated in the most diluted wells were again subjected to dilution cloning as above. The results showed that four kinds of ameboid organism (based on cellular shapes determined by microscopy) dominated in many wells independently, but one grew very slowly. We isolated clones of the other three kinds and used them for further analyses.

To identify the lineages to which the three ameboid organisms belonged, 18S rRNA genes of the three species were amplified by PCR with the primers MoonA and MoonB[53]. The amplified DNA was cloned into the pGEM-T Easy vector (Promega) and sequenced. The 18S rDNA sequences were subjected to a BLASTn search against the non-redundant nucleotide sequence database of the NCBI.

**Preparation of bacterial prey**. *S. elongatus* was cultured in BG-11 liquid medium in Erlenmeyer flasks at 30 °C under continuous light ($20 \mu E\, m^{-2}\, s^{-1}$) on a rotary shaker. To prepare the pale *S. elongatus* in which chlorophyll *a* and phycobilin levels are reduced, this species was cultured under nitrogen-depleted conditions[18]

as follows. Exponentially growing cells in BG-11 were harvested by centrifugation at $2000 \times g$ for 15 min and then washed twice with nitrogen-depleted BG-11 (BG-$11^{-N}$). The washed cells were resuspended in 40 mL of BG-$11^{-N}$ to give an optical density at 750 nm ($OD_{750}$) of 0.25 and were cultured in 100 mL test tubes at 30 °C under illumination ($55 \mu E\, m^{-2}\, s^{-1}$) with aeration (0.3 L ambient air/min) for 7 days.

Heme, which is a complex of iron ion and porphyrin, is also known to generate ROS[54]. To subtract the effects of heme from chlorophyll *a*-stained *E. coli*, an *E. coli hemA* deletion mutant ($\Delta hemA$)[55] was stained with chlorophyll *a* as follows. *E. coli* $\Delta hemA$ was obtained from the National Bio Resource Project. *E. coli* W3110 (the parental strain of $\Delta hemA$) and $\Delta hemA$ were cultured in Luria broth (LB) liquid medium in Erlenmeyer flasks at 37 °C on a rotary shaker. For $\Delta hemA$, the medium was supplemented with 50 mM sodium pyruvate[55]. To stain $\Delta hemA$ cells with chlorophyll *a*, $5 \times 10^8$ cells in 80 μL of distilled water were mixed with 400 μL of chlorophyll *a* (034–21361; Wako, Japan) dissolved in acetone (3.0 μL of 10 mg/mL chlorophyll *a* in DMSO was diluted with 397 μL of acetone). Then, acetone was evaporated in a centrifugal evaporator (CVE-3100; EYELA, Japan) at 1000 rpm and 30 °C for 2 h. The chlorophyll-stained cells in the residual water were washed with 1/16-strength BG-11 (BG-$11^{-16}$) and resuspended in BG-$11^{-16}$. As a negative control, $\Delta hemA$ cells were processed as above except that the chlorophyll *a* solution was not added. Just before feeding the *Naegleria* sp. the bacterial prey prepared as above, the prey were washed with BG-$11^{-16}$ three times.

To quantify chlorophyll *a* in bacterial prey, cell pellets from a 1.0 mL culture at $OD_{600} = 1.0$ for *E. coli* or $OD_{750} = 1.0$ for *S. elongatus* were resuspended in 1.0 mL of methanol. Cells were removed by centrifugation at $10,000 \times g$ for 10 min.

Absorbance of the supernatant fractions at 665 nm was measured with a spectrophotometer[56]. To quantify phycocyanins in cyanobacterial prey, absorbance of the respective cultures at 678 and 620 nm was measured[57] using a spectrophotometer (UV-2600; Shimadzu) with an integrated sphere attachment (ISR-2600PLUS; Shimadzu).

**Cultivation of the three excavate and amoebozoan species**. To maintain the amoebozoan and excavate cultures, respective cells were co-cultured with *E. coli* W3110 prey (grown in LB) in BG-11$^{-16}$ on a Petri dish at 20 °C under illumination (20 $\mu$E m$^{-2}$ s$^{-1}$). For long-term preservation, the cells were suspended in BG-11 supplemented with 10% (v/v) DMSO (*Naegleria* sp. and *Acanthamoeba* sp.) or 10% (v/v) glycerol (*Vannella* sp.) and preserved at −80 °C.

To culture the excavate and amoebozoans in a synthetic organic medium without bacterial prey, 1/10 strength Proteose Peptone Glucose medium (PPG$^{-10}$; Culture Collection of Algae and Protozoa; https://www.ccap.ac.uk/media/documents/PPG.pdf) was used. *Naegleria* sp. and *Acanthamoeba* sp., but not *Vannella* sp., were able to grow in PPG$^{-10}$ supplemented with 50 nM kanamycin at 20 °C.

In the co-cultivation system that was used for the experiments, the excavate and amoebozoans and bacterial prey were co-cultivated in Petri dishes (35, 50, or 84.5 mm in diameter or a 24-well plate). The dishes were put on a transparent acrylic box (35 cm width × 25 cm depth × 5 cm height) in which water at 20 °C was circulated (Fig. 1g). Dishes were illuminated from the bottom of the acryl box by fluorescent lamps and aluminum foil was used to shade the dishes for the incubation in the dark (Fig. 1g). The system was constructed in a temperature-controlled growth chamber. When the co-culture was performed at 20 °C, the chamber was set at 20 °C. When the co-culture was performed at 25 °C, the chamber was set at 31.5 °C. Via the balance between the temperatures of 31.5 °C of the chamber and 20 °C of the circulating water, we confirmed that the medium in the dishes reached 25 °C. Because the feeding and growth rates at 25 °C were higher than those at 20 °C, 25 °C was applied for the assays for growth and phagocytosis of the excavate *Naegleria* sp.

**Quantification of *Naegleria* sp. growth**. *Naegleria* sp. was pre-cultured with green (OD$_{750}$ = 0.4) or pale (OD$_{750}$ = 0.3) *S. elongatus* prey in 13 mL of BG-11$^{-16}$ in 84.5 mm Petri dishes at 25 °C in the dark for 2 h. (Because the feeding and growth rates at 25 °C were higher than those at 20 °C, 25 °C was applied in this assay.) To accelerate the ingestion of bacterial prey by *Naegleria* sp. cells, *E. coli* W3110 (OD$_{600}$ = 0.1) was also added to the respective co-culture. After the pre-culture, *Naegleria* sp. cell density in the Petri dish was determined. Then, a portion of the co-culture was transferred into 35 mm Petri dishes to give a density of 400 *Naegleria* sp. cells/mm$^2$ and then incubated for 40 min in the dark to let *Naegleria* sp. cells adhere to the bottom of the Petri dishes. The liquid medium and free prey were removed by gentle rinsing with BG-11$^{-16}$. Then, 3 mL of BG-11$^{-16}$ supplemented with green ($4.0 \times 10^7$ cells/dish) or pale ($1.0 \times 10^8$ cells/dish; the difference in the concentrations between green and pale is because pale was consumed by *Naegleria* sp. faster than green) *S. elongatus* and *E. coli* ($5.0 \times 10^8$ cells/dish) was added to *Naegleria* sp. culture in the Petri dishes. The co-cultures were incubated at 25 °C in the dark for 1 h and then (0 min) kept in the dark or transferred to low-light (200 $\mu$E m$^{-2}$ s$^{-1}$) or high-light conditions (500 $\mu$E m$^{-2}$ s$^{-1}$). Micrographs were taken at 0, 90, 180, and 360 min, and the density of *Naegleria* sp. cells per area was determined.

To evaluate the effect of green or pale *S. elongatus* prey that was ingested by *Naegleria* sp. on *Naegleria* sp. growth, free bacterial prey outside the *Naegleria* sp. cells was removed by gentle rinsing of the dish with fresh BG-11$^{-16}$ medium after 1 h co-culture in the dark. The *Naegleria* sp. cells in fresh medium were placed between the bottom of the dish and a cover glass with a ~3 mm spacer to prevent dying (round) *Naegleria* sp. cells from floating away. Then, the *Naegleria* sp. cells were exposed to high light for 60 min. To observe the effect of H$_2$O$_2$ on *Naegleria* sp. cells, free bacterial prey (*E. coli* W3110) was removed by gentle rinsing of the dish with fresh BG-11$^{-16}$ medium and then H$_2$O$_2$ was added to the *Naegleria* sp. culture to give a concentration of 0.8 mM. Immediately after the addition of H$_2$O$_2$, *Naegleria* sp. cells were incubated under a microscope.

**RNA preparation for transcriptomic analyses**. The excavate *Naegleria* sp. and the amoebozoans *Acanthamoeba* sp. and *Vannella* sp. were used as predators. Four types of bacteria, green and pale *S. elongatus* and *E. coli* Δ*hemA* with or without chlorophyll *a* staining (both were treated with acetone), were used as prey. The amoebozoan or excavate cells ($9.8 \times 10^5$ per 50 mm Petri dish; 500 cells/mm$^2$) were co-cultured with green or pale *S. elongatus* ($3.0 \times 10^8$ cells per 50 mm Petri dish; 300 cells/amoebozoan or excavate cell) or *E. coli* with or without chlorophyll *a* staining ($1.0 \times 10^9$ cells per 50 mm Petri dish; 1000 cells/amoebozoan or excavate cell). The *E. coli* concentration was higher than that of *S. elongatus* because the excavate and amoebozoans consumed *E. coli* prey faster than *S. elongatus* prey. The co-culture was incubated at 20 °C in the dark for 12 h and then illuminated (200 $\mu$E m$^{-2}$ s$^{-1}$) for 1 h. Total RNA of the excavate and amoebozoans was extracted from the cultures just before (dark) or 1 h after the light illumination.

*Naegleria* sp. and *Acanthamoeba* sp. were also cultured without bacterial prey in organic PPG$^{-10}$ supplemented with 50 $\mu$g/mL kanamycin. *Naegleria* sp. or

*Acanthamoeba* sp. ($9.8 \times 10^5$ cells per 50 mm Petri dish; 500 cells/mm$^2$ at the culture onset) was cultured in the dark for 12 h and either illuminated (200 $\mu$E m$^{-2}$ s$^{-1}$) or incubated in the dark with 0.1 mM H$_2$O$_2$ for 1 h. The cells just before illumination or the addition of H$_2$O$_2$ were used as negative controls. For the RB treatment, 30 nM RB was added to the culture at the onset of 12 h of incubation in the dark, followed by illumination for 1 h. The cells just before illumination were used as a negative control. The H$_2$O$_2$ and RB treatments were applied only to *Naegleria* sp.

To extract total RNA from the excavate and amoebozoans, liquid medium and bacterial prey were removed as much as possible by gentle washing with fresh medium and the excavate and amoebozoans adhering to the bottom of the Petri dish were resuspended with 400 $\mu$L of TRIzol reagent (Invitrogen). The total RNA was extracted using the RNeasy Mini Kit (Qiagen).

Regarding the number of replicates, for some culture conditions, we were able to obtain three independent replicates of RNA-seq data with the predators newly prepared from the original frozen stock and bacterial prey prepared at the time of use (Supplementary Fig. 3). However, for other conditions, we were able to obtain only a single or two replicates of RNA-seq data because, as mentioned above, the growth of *Acanthamoeba* sp. and *Vannella* sp. became unstable after long-term storage (Supplementary Fig. 3). The statistical analyses were applied only to cases in which three independent replicates of data were obtained.

Cultivation of *Naegleria* sp. and *Acanthamoeba* sp. with green *S. elongatus* prey was performed three times independently on different days for three independent replicates of RNA-seq analyses. In a similar manner, cultivation of the amoebozoan *Vannella* sp. with green *S. elongatus* prey was performed two times independently on different days. For *Naegleria* sp., cultivations under other conditions (i.e., with pale *S. elongatus* prey, without prey in the organic medium, H$_2$O$_2$ or RB treatment, with *E. coli* stained or unstained with chlorophyll *a*) were also performed three times independently. Other cultures (*Acanthamoeba* sp. with pale *S. elongatus* prey or without prey in the organic medium; *Vannella* sp. with pale *S. elongatus* prey) were performed once. In each replicate, samples to be compared were prepared from cultures at the same time (i.e., groups 1, 2, and 3 of each predator species in Supplementary Fig. 3).

**RNA-seq analyses**. mRNA of the respective samples was purified from 0.6 to 14.8 $\mu$g of total RNA with Dynabeads Oligo(dT)25 (Life Technologies). The purified mRNA was fragmented into small pieces using divalent cations under elevated temperature. The cleaved RNA fragments were used for first-strand cDNA synthesis using SuperScript II Reverse Transcriptase (Invitrogen) and random primers. Then, second-strand cDNA synthesis was conducted. These cDNA fragments were then subjected to an end repair process and the ligation of adapters. These products were purified and enriched by PCR to create the final cDNA library. Sequencing was performed by HiSeq 2500 with a 100-bp end format with the TruSeq SBS kit ver. 3 (Illumina). The reads were cleaned up using the cutadapt program ver. 1.81[58] by trimming low-quality ends (<QV30) and adapter sequences and by discarding reads shorter than 50 bp.

For de novo assembly of the RNA-seq reads, reads from *S. elongatus* and *E. coli* (W3110) prey were removed by Bowtie2 ver. 2.1.0[59]. The de novo assembly of the excavate *Naegleria* sp. and the amoebozoans *Acanthamoeba* sp. and *Vannella* sp. reads was conducted by Trinity ver. 2.0.6[60] of the DDBJ pipeline[61,62] with the paired-end mode and the option --min_contig_length 200. When splicing variants of a gene were found, the longest transcript was selected as a representative mRNA sequence.

The assembled mRNA contigs were subjected to BLASTx and BLASTp (query mRNA sequence was translated into amino acid sequences using a TransDecoder ver. 2.0.1; http://transdecoder.github.io) search against the Uniprot Swiss-Prot protein database. The results of BLAST searches were processed by the Trinotate ver. 2.0.2 comprehensive annotation suite (https://trinotate.github.io) to produce the annotation database of mRNA contigs.

**Comparison of mRNA levels between species and conditions**. The RNA-seq reads were mapped to the de novo assembled mRNA contigs of respective excavate and amoebozoans by Bowtie2 ver. 2.1.0[59]. Although the number of reads per replicate in a certain species and culture conditions varied because the culture and RNA extraction of respective replicates were performed independently on different days, the difference in number of mapped reads in each comparative pair (i.e., light vs. dark; with vs. without H$_2$O$_2$) was less than twofold except for RNA-seq data of *Vannella* sp. with pale *S. elongatus* prey (light/dark = 2.58) (Supplementary Table 2). The raw count data and FPKM values of respective contigs under respective conditions were calculated by RSEM ver. 1.2.21[63].

The count data were analyzed in R using edgeR ver. 3.21.1[21]. To cut the contigs assigned with low RNA-seq reads, the contigs were analyzed only when total count per million of two comparative conditions (i.e., light vs. dark; with vs. without H$_2$O$_2$ in the dark) was > 5.0. Normalized count data of two comparative conditions in each replicate were analyzed as a paired dataset with a generalized linear model fit because each biological replicate was performed independently on a different day. Up- and downregulated genes were defined as having an FDR < 0.01 and a log2-fold change (light vs. dark; with vs. without H$_2$O$_2$ in the dark) > 0, and FDR < 0.01 and a log2-fold change < 0, respectively.

Among significantly up- and downregulated genes under chlorophyll and RB treatments upon illumination, to distinguish the effect of chlorophyll/oxidative stress from the effect of light, the contigs were included in the Venn diagram in Fig. 4a only when the *p*-value was < 0.05 in a paired and one-tailed *t*-test between (i) log2-fold change (light/dark) of 'Chl' vs. that of 'w/o Chl', and (ii) log2-fold change of 'RB' vs. that of 'w/o prey'. Thus, in Fig. 4a, the upregulation (or downregulation) by a chlorophyll treatment was defined by being more upregulated (or downregulated) upon illumination in cells feeding on chlorophyll-stained *E. coli*, respectively, than those feeding on unstained *E. coli* upon illumination. In a similar manner, the upregulation by RB was defined by being more upregulated (or downregulation) upon illumination in cells with RB than those without RB.

To define orthogroups between *Naegleria* sp. and *Acanthamoeba* sp. the predicted protein-coding sequences of *Naegleria* sp. and *Acanthamoeba* sp. were inferred using by OrthoFinder ver. 2.3.3[22] with default parameters. To compare the patterns of transcriptomic change upon illumination between the two species, the values of logFC, which were calculated from comparison of light/dark ratio in *Naegleria* sp. and *Acanthamoeba* sp. feeding on green *S. elongatus* prey by edgeR, were used (contigs assigned with low RNA-seq reads were omitted as described above). The values of logFC were averaged when two or more contigs of a species were included in an orthogroup. Spearman rank correlation coefficient was calculated with R.

**t-SNE analysis of transcriptomic changes in *Naegleria* sp.** For t-SNE analyses, the contigs assigned with low RNA-seq reads were omitted as described above. Then the raw count data were normalized together using TCC package. The log2-fold changes in respective datasets in respective culturing conditions (light vs. dark or with or without $H_2O_2$ in the dark) were analyzed by t-SNE[27] (Rtsne package: seed 1, perplexity = 6, pca = F, iteration = 10,000, theta = 0.0).

**KEGG and GO enrichment analyses**. The up- and downregulated contigs (FDR < 0.01) of *Naegleria* sp. and *Acanthamoeba* sp. fed on green *S. elongatus* prey were categorized according to the second-level terms without redundancy within the categories in the KEGG classification. The KEGG Orthology ID assignment was performed for the assembled contigs of *Naegleria* sp. and *Acanthamoeba* sp. through the KAAS pipeline[23]. The GO enrichment analysis was performed for the up- and downregulated genes of *Naegleria* sp. and *Acanthamoeba* sp. fed on green *S. elongatus* prey by GOseq ver. 1.22.0[64]. By using the GO terms in the Trinotate annotation, terms that were enriched in contigs that were up- or downregulated upon illumination (*P* < 0.05) were determined by GOseq.

**Quantitative RT-PCR**. Quantitative RT-PCR was performed with a 1/500 aliquot of cDNA prepared from 5 μg of total RNA for each *Naegleria* sp. sample using the Power SYBR Green PCR Master Mix (Applied Biosystems), using a StepOnePlus Real-Time PCR system (Life Technologies) and a 20-μL reaction mixture with the primers listed in Supplementary Table 7. *Elongation Factor 1 alpha* (*EF1α*) (comp225284_c0_seq1|m.13820) was used as an internal control.

**Quantification of phagocytic activity**. *Naegleria* sp. (500 cells/mm²) was co-cultured with green or pale *S. elongatus* prey ($6.0 \times 10^4$ cells/mm²) in 1 mL of BG-11$^{-16}$ in a well of a 24-well culture plate (surface area of each well was 1.86 cm²) at 25 °C for 4 h in the dark. (Because the feeding and growth rates at 25 °C were higher than those at 20 °C, 25 °C was applied in this assay.) Then, the plate was kept in the dark or transferred to light (200 μE m$^{-2}$ s$^{-1}$) conditions and further incubated for 1 h. Then, fluorescent beads of 1.0 μm in diameter (Fluoresbrite® YG Microspheres 1.00 μm; Polysciences, Inc.) were added to the culture to give a concentration of 8 beads/mm² and further incubated for 1 h under dark or light conditions. The number of beads that were ingested by *Naegleria* sp. cells was then determined.

**Quantification of the digestion rate**. To label the green and pale *S. elongatus* prey with fluorescence, the cells were suspended in 25 μg/mL FM1–43 (Invitrogen) dissolved in BG-11$^{-16}$ and incubated at room temperature in the dark for 12 h. Then, the stained prey was washed three times with BG-11$^{-16}$.

*Naegleria* sp. (1500 cells/mm²) was co-cultured with FM1–43-stained prey ($9.0 \times 10^4$ cells/mm²) in 1 mL of BG-11$^{-16}$ in a well of 24-well culture plates at 25 °C for 4 h in the dark for *Naegleria* sp. cells to ingest the prey. Then, the liquid medium and free bacterial prey were removed from the well as much as possible by gentle rinsing with BG-11$^{-16}$, and *Naegleria* sp. adhering to the bottom of the plate was resuspended in 300 μL of BG-11$^{-16}$. Then, the samples were put on MAS-coated slide glass (#S9115, Matsunami) and incubated at 25 °C in the dark for 30 min to immobilize the *Naegleria* sp. cells on the glass and inhibit them from continuing to ingest bacterial prey. Then (hour 0), the glass on which *Naegleria* sp. cells was immobilized was incubated in a Petri dish with moist filter paper at 25 °C under dark or light (200 μE m$^{-2}$ s$^{-1}$) conditions. Micrographs by fluorescence microscopy were taken every hour from hour 0 to 4. The fluorescence intensity of prey that in *Naegleria* sp. cells (per area of the cells) and outside the cells (as a control) (per area) was determined with ImageJ software[65].

**Protein domain predictions and phylogenetic analyses**. Putative transmembrane domains were predicted by TMHMM 2.0 (http://www.cbs.dtu.dk/services/TMHMM/), and Rieske [2Fe-2S] iron-sulfur and PAO-like domains were predicted by Pfam 32.0 (https://pfam.xfam.org). For phylogenetic analyses, deduced amino acid sequences of proteins that possess both Rieske [2Fe-2S] iron-sulfur domain and PAO-like domain were collected by BLASTp searches in the National Center for Biotechnology Information (NCBI) database (Accession numbers or locus IDs are indicated in Fig. 6a). The sequences were aligned by Clustal X 2.0[66] and manually refined, and 242 amino acid residues were used for the phylogenetic analyses. Maximum-likelihood trees were constructed using RaxML 7.0.4[67] with 100 replicates using the WAG matrix of amino acid replacements assuming a proportion of invariant positions and four gamma-distributed rates (WAG + I + gamma model). Bayesian inference was performed with the program MrBayes version 3.1.2[68] using the WAG + I + gamma model. For the MrBayes consensus trees, 1,000,000 generations were completed with trees collected every 100 generations.

**Reporting summary**. Further information on research design is available in the Nature Research Reporting Summary linked to this article.

## Data availability

All relevant data supporting the key findings of this study are available within the article and its Supplementary Information files or from the corresponding author upon reasonable request. The 18S rDNA sequences of the excavate *Naegleria* sp. and the amoebozoans *Acanthamoeba* sp. and *Vannella* sp. were deposited at DDBJ (accession numbers LC368140, LC368141, and LC368142). All of the RNA-seq raw data obtained in this study and assembled contig sequences were deposited at the DDBJ Sequence Read Archive (accession codes BioProject PRJDB6846 for *Naegleria* sp., BioProject PRJDB6847 for *Acanthamoeba* sp., and BioProject PRJDB6848 for *Vannella* sp., respectively) and at the DDBJ Transcriptome Shotgun Assembly Database (accession codes IADA00000000 for *Naegleria* sp., IACY00000000 for *Acanthamoeba* sp., and IACZ00000000 for *Vannella* sp.). The source data underlying Figs. 1e, 2a, 4b, 5b, and 5d and Supplementary Figs. 1, 2, and 6 are provided as a Source Data file.

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

## Acknowledgements

We thank Dr. Kashiyama in Fukui University of Technology for useful suggestions and members of the S.-y.M. laboratory for their technical advice and support. Computations were partially performed on the National Institute of Genetics (NIG) supercomputer at the Research Organization of Information and Systems of the NIG. This study was supported by Ministry of Education, Culture, Sports, Science and Technology of Japan grants to S.-y.M. (16K14791 and 17H01446) and to A.U. (17J08575), and by the Ministry of Education, Culture, Sports, Science and Technology-supported Program for Strategic Research Foundation at Private Universities to Yu.K. and H.Y. (2013–2017 S1311017).

## Author contributions

A.U. and S.-y.M. conceived and designed the experiments. A.U., T.F., and S.-y.M. isolated the excavate and amoebozoans. A.U. and Yus.K. cultured the excavate and amoebozoans under several conditions and performed experiments. R.O., Yu.K., and H.Y. performed high-throughput sequencing. A.U., S.H., and R.O. analyzed RNA-seq data. S.-y.M. conducted phylogenetic analyses. A.U. and S.-y.M. wrote the paper.

## Competing interests

The authors declare no competing interests.
