## [Peer Review File · Nature Communications]

Reviewers' Comments:

Reviewer #1:

Remarks to the Author:

The study presented examines the effects of photosynthetic prey on heterotrophic amoeba. The authors show that photosynthetic activity of the prey affects gene expression profiles of evolutionary distantly related amoeba species in a similar way. These transcriptomic changes are further accompanied by a reduction in phagocytotic uptake and an increased rate of digestion of engulfed prey, suggesting an active reduction of phototrophs inside the predator cells to reduce oxidative stress caused by the prey. These effects are supposedly similar to observations made in facultative symbiotic associations. The authors therefore conclude that feeding on autotrophic prey requires evolutionary adaptations of the predator to cope with prey induced oxidative stress, which might be conducive to the evolution of symbiotic relationships with autotrophic symbionts.

The manuscript is well written and easy to understand. I generally agree with the underlying hypothesis and find it quite compelling, However, I do have strong reservations regarding the experiments and analyses presented and the extent to which they provide conclusive evidence for the findings.

The transcriptome analysis presented, for instance, is way below the standard I would expect for a RNA-seq analysis. The current manuscript is not clear on how the gene expression analyses were conducted since important information is missing, but it is clear that the level of replication is too low (only 2 replicates per treatment). Further, I find that the differential gene expression analysis has not been performed using standard normalization and statistical analysis methods, but rather that differentially expressed genes were determined on the level of FPKM values and fold change instead. Furthermore, no additional validation of the contigs was performed using qPCR. With regard to RNA-seq analysis there're several other problems. I'd recommend performing the experiment again using at least 3 biological replicates per treatment and any of the standard analysis pipelines using raw counts for instance (e.g. Trinity, edgeR, kallisto, etc.). Interestingly, it appears that the authors actually used edgeR to determine enriched GO categories.

Furthermore I'd expect a qPCR validation of the genes identified to respond to the treatment (e.g. myosin, Glutathione peroxidase, etc.) as independent validation of the RNA-seq results as mentioned above. The material and methods part should further provide information on the number of mapped reads per replicate and more detailed information on the way the experiment was actually performed. Were experiments performed on different batches to control for potential batch effects? At what time points were the different samples taken? This is important to assess if similarities in expression profiles might stem from sampling time point biases i.e. samples taken at the same time point might show higher similarity simply because they were taken at the same time.

Similarly, I cannot find important information on the other experiments. How many replicates were used per treatment in the different feeding experiments? The manuscript provides this information for some experiments (e.g. Fig. 2 A) but not all.

As for the results presented, the authors state that expression profiles are similar between the different amoeba strains; however, I cannot find an analysis showing this (e.g. PCA, overlap of differentially expressed genes, etc.). Instead the author present differentially expressed genes of selected GO categories and a Venn diagram without corresponding numbers for the overlap. Based on the list of enriched GO terms listed in Supp. Table 1-6 looks like GO terms might have been "cherry picked" i.e. it seems a bit as if the authors picked a handful of terms out of a quite extensive list. Looking at the supplementary data providing the full list of GO terms (Supp. Table 1-6) further substantiates this

impression.

With regard to the effects of ROS, the authors have used H₂O₂ as comparison, for instance, but no experiment has been performed to show that ROS levels are indeed elevated when photosynthetic prey is ingested and that ROS levels are reduced by the, supposedly, increased digestion rate. Several dyes exist to perform in vivo staining and measurement of ROS levels (e.g. CellROX) and I suggest including such an analysis.

Finally, I'd like to comment on the comparisons to endosymbiotic associations. It is not clear if ROS is indeed the trigger for symbiont expulsion or degradation or if ROS is rather an indicator of dysfunctional metabolic interactions i.e. photoinhibition leading to reduced carbon transfer rates and dysfunctional metabolic regulation of the symbiotic relationship. While I agree that the phenotype (reduction of endosymbionts in the tissue) is similar, it cannot be excluded that the underlying cause and mechanism is different. I think it would take additional experiments to show that these processes are indeed comparable.

In summary, while I do find the hypothesis presented interesting and compelling, I do not feel that the results presented provide sufficiently strong evidence given the severe experimental and analytical flaws.

Reviewer #2:

Remarks to the Author:

This study aims to prove that when a predator is fed with photosynthetic prey under illumination, the photosynthetic oxidative stress of the prey is effecting the metabolic activity of the predator. This consideration could improve our understandings of the evolutionary establishment of photosynthetic eukaryotes.

The finding suggested in this study are, in my view, novel. Reading this study has been very fascinating. The Figure 5 proposes a very interesting model. If published this study will attract much interest.

Before publication, in my view, the authors should be able to prove the following points.

i) predator without prey does not grow (R0)

ii) predator + dead* photosynthetic prey, in the dark can grow at a certain rate (R1)

iii) predator + alive photosynthetic prey, in the dark can grow at a certain rate (R2)

With R1 and R2 should be similar

iv) predator + dead* photosynthetic prey, in the light (low**) can grow at a certain rate (R3)

v) predator + alive photosynthetic prey, in the light (low**) can grow at a certain rate (R4)

With R4 should be > R3

With R3 should be similar to R2 and R1

vi) predator + dead* photosynthetic prey, in the light (high**) can grow at a certain rate (R5)

vii) predator + alive photosynthetic prey, in the light (high**) can grow at a certain rate (R6)

With R4 should be > R6

With R5 should be similar to R3, R2 and R1

*non photosynthetically active

**low light should be able to power the photosynthesis of prey without generation of oxidative stress.

High light should be able to power the photosynthesis of prey but with generation of oxidative stress

Figure 2A, to better compare those three conditions (dark, low light and high light) please make the Y axis identical.

Please provide the relative negative control (therefore Naegleria without feeding). Also, please provide three equal graph showing the cell growth of *S. elongatus* without the Naegleria. Could the authors provide some data about the amount of oxygen in the three conditions (dark, low light and high light)?

I am not sure about the use of *E. coli* cells incubated in acetone with (Chl) or without (w/o Chl) chlorophyll a, dried, and then rehydrated. I do not expect the Chl transferred inside the *E. coli* able to do any photochemistry. Autofluorescence is OK, but not photochemistry.

Figure 4. Could the author explain why the decrease in FM1-43 fluorescence should be a proxy for digestion rate? And why this rate should be related with the presence of light.

Paolo Bombelli

Reviewer #3:

Remarks to the Author:

The authors present an exciting study exploring the physiological response of amoeba when feeding on photosynthetic prokaryotes. The authors explored this question in a very thorough and careful way, testing various hypothesis and subsidiary questions. The authors developed clever experiments such as the culture with fluorescent beads to practically quantify food uptake. This helps to produce a manuscript not only based on transcriptomic analysis but on empirical data as well. This work opens interesting perspectives on the evolution of photosynthesis in eukaryotes through endosymbiosis. The English and writing is in general good. However it is sometimes hard to follow the logic of the authors as they present results in a little confused way. Notably, I suggest that the discussion on the transcriptomic analysis should be entirely rewritten for a better understanding.

Major comments:

Page 6 line 104-119: The authors studied the growth rate of amoeba under different culture conditions. However the authors did not actually calculated a growth rate. The authors should present a statistical analysis of their results. This might change the discussion. For example, based on the histograms presented on the Figure 2, I am not convinced that the growth rates between the "Dark" and "Low light" are similar each other. It seems that even in "Low light" the growth rate is stagnant for the first 90 minutes. Statistical comparison of the different rates could help to more clearly understand the reaction of amoeba to the culture conditions. The author should also present all the histograms using the same y-axis in order to ease the visual comparison between the different cultures.

Page 7 line 134-137: This sentence is confusing and could be rewritten. The authors are explaining how they searched for "false positive" transcripts identified from their experiment.

Page 7-9 line 121-175: The transcriptome analysis the authors present is impressive for the control experiments that the authors conducted to explore further the expression regulation occurring during feeding of their amoeba isolates. However the different analysis and controls conducted by the authors are presented in a somewhat disordered way and it is hard to understand what set of the transcripts the authors are discussing. I would suggest that the authors more clearly separate the analysis in two different parts. First, the main analysis of the up/down regulated transcripts and exploring the main GO terms represented and species specific regulations. And in a second part, discuss those results in the light of the different control experiments. I think this would help for readers of the manuscript without altering the main results of this transcriptomic analysis (i.e. expression regulation is altered while feeding on photosynthetic prey but most of those regulation can be triggered by oxidative-stress

and/or illumination). Rewriting of this part of the manuscript should also be accompanied by a revision of the Figure 3B. The Venn diagrams are not self-explanatory in my opinion and the numbers presented do not add up with the Up/Down regulated transcripts from the main experiment (i.e. feeding on photosynthetic prey under low light).

Finally, a more thorough discussion on the evolution of those regulations would be a nice addition to this manuscript. The authors could for example explore differentially regulated transcripts without annotation that are shared among the three species. Despite not having an annotation exploring the domain encoded and those transcripts could bring interesting evolutionary insights and suggest target for further studies. But this might not be statistically meaningful (see below).

Page 21 line 418-421: The entire transcriptomic analysis are based on duplicate and not triplicate. It is generally accepted that triplication is the minimum for statistically reliable transcriptomic analysis (although triplicates are actually still not sufficient, see <https://www.ncbi.nlm.nih.gov/pmc/articles/PMC4878611/> for a discussion on this). Therefore I am not sure that the transcriptomic analysis, despite being well designed and interesting (see above), are statistically meaningful. However this does not constitute a reason to reject the paper in my opinion as the most interesting results (again in my opinion) are based on empirical observation (e.g. feeding rate) and globally are in accordance with the main trends observed in the transcriptomes.

Minor comments:

Page 3 line 32: References concerning the dates advanced by the authors should be added.

Page 3 line 48: The term "transparent" might be not suited. I suggest the authors use the term colorless when referring to non-photosynthetic organisms.

Page 4 line 63: "suggesting that the responses" could be changed "suggesting that these responses".

Page 4 line 69: "which is observed".

Page 5 line 82: The authors used a blastn search to identify the amoeba species they isolated. The best hits scores should be presented in a supplementary table. Furthermore the authors could present a 18S rRNA phylogenetic tree including their isolates to determine more clearly their identification.

Page 5 line 87 Page 7 line 121 Page 13 line 255: "evolutionarily distantly related" could be replaced simply by "distantly related" or "evolutionarily distant".

Page 8 line 154-155: "but not in unstained E. coli upon illumination" .

Page 10 line 186-191: I think cox15 is also a component of the cytochrome c oxidase assembly (at least in yeast) and its up-regulation could have an effect on mitochondrial respiration as well. The authors should explore on this.

Reviewer #4:

Remarks to the Author:

The report presents an interesting study about the cellular and molecular responses of three different types of predatory protists after engulfing cyanobacterial preys. The main hypothesis is that the protist cellular systems that cope with oxidative stress have an important role contending with the production of reactive oxygen species (ROS) by the photosynthetic cyanobacterial preys. The working hypothesis has evolutionary implications, because a widely accepted scenario posits that the origin of primary photosynthetic organelles (i.e., plastids) likely involved similar ecological interactions between predatory protists and cyanobacterial preys. It has been discussed largely the importance of the prey survival mechanisms in the establishment of long term endosymbiotic associations, but this contribution of Uzuka and collaborators focuses on the other side of the coin: how do the unicellular host copes with the putative toxicity, given the ROS release, of the photosynthetic partner?

Uzuka and collaborators designed a series of experiments to evaluate at different levels the responses of predatory protists to different intensities of photosynthetic activity in their cyanobacterial preys. The authors suggest that in the different studied protists similar gene expression responses occur to

regulate the phagotrophic activity, prey digestion and mechanisms of protection against oxidative stress. It is suggested also that these gene expression changes are the predator responses to limit the potential damage caused by the cyanobacteria-produced ROS, to reduce the capture of more cyanobacteria and to simultaneously accelerate the digestion of the captured photosynthetic cells.

Overall the study presents experimental results that have important implications to better understand cellular mechanisms involved on early stages of endosymbiotic associations with photosynthetic partners. However, I have a series of concerns that in my view have to be addressed before consideration for publication in Nature Communications.

Major concerns

1. The Introduction section should be re-written using more precise ideas and rigorous organization. The second paragraph uses loose sentences to describe an important piece of the context: the general hypothesis about the origin of primary plastids. Please expand this point and describe the different ideas detailed in plenty of recent publications. For example, the opening sentence "It is believed that chloroplasts were established through predation and the temporary retention of photosynthetic prey" is an oversimplification of the sophisticated scenarios that have been proposed, which are relevant to justify both the tested hypothesis and the design of the experimental work. Then, the use of the "herbivorous" adjective is odd for protists that actually are referred in the text as consumers of cyanobacteria, which are not plants or herbs. The third paragraph includes mostly description of results (lines 58-66) and sentences that seem more adequate for discussion (Lines 67-72). Please revise the entire introduction section.
2. Do the three studied protist species regularly consume *Synechococcus elongatus* in their natural habitat? It is mentioned, and illustrated, that the sampling site is dominated by 'microalgae' (photosynthetic eukaryotes). Is there any specific information about the presence of *Synechococcus elongatus* in the same water body? Why did you use cyanobacteria instead of the apparently abundant algal preys? Clarify and justify your decision.
3. How did you establish the time of exposure to bacterial preys in the co-cultivation experiments (Figure 2)? Please explain.
4. It is likely that the captured cyanobacteria are photosynthetically active under light conditions, but it will be important to evaluate the actual carbon fixation rate directly because that aspect is another important part of the general hypothesis of plastid origins by endosymbiosis: the prey likely releases fixed carbon compounds into the host/predator cytoplasm and that also must play a key role in the digestion response of the host. Is there any active or passive transport of carbon compounds from the cyanobacteria to the protist cell? Is this different in "green" and "pale" prey?
5. Did you try to expose protist to "green" cyanobacterial preys under light conditions in the presence of inhibitors of the photosynthesis or compounds that limit ROS formation by the photosynthetic electron chain? This would be a good experimental control to untangle the set of protist genes expressed exclusively in response the ROS production by the cyanobacterial photosynthetic activity.
6. I do not understand clearly why 'pale' cyanobacterial preys were prepared. It is formally possible that the actual endosymbiosis that gave rise to primary plastids occurred with 'pale' endosymbionts under low light if the nutrient flux between both partners allowed a stable endosymbiotic association. Under such scenario, the level of ROS production by the cyanobacteria would not have the same impact. Is that the reason why you decided to include 'pale' cyanobacterial preys in your experiments?

Please explain the decision in more detail.

7. Figures 2 and 3 are unnecessary busy. In my opinion the number of panels is excessive. Is this because a limitation in the number of allowed figures? The figures are hard to follow and I strongly recommend more concise streamlined versions. For example, panels A, B, C and G of Figure 2 are in my opinion superfluous.

Minor comments

8. Use 'plastids' (a generic term) instead of 'chloroplasts' (i.e., the plastids of green algae and plants).

9. 'Amoeba' is a loose term with no phylogenetic significance. I would suggest you use 'excavate' or 'amoebozoan', depending on the species, instead of 'amoeba'.

10. Line 87- "evolutionally distantly related' is a vague sentence. Please be more specific.

11. Line 112- Why is the description of the cells shape relevant?

12. Lines 115-116. This is an interesting but speculative sentence. You need to provide evidence of that.

13. Lines 235-236. "This difference is probably because the pale prey, which was prepared by nitrogen starvation, was more easily digested by amoebae". What is the evidence to suggest that? Overall, the text contains several speculative sentences that have to be reduced to a minimum, or eliminated.

14. Lines 260-263 – This is a deterministic statement. All free-living cells, predatory or not, have mechanisms to cope with oxidative stress, but those are not 'prerequisites' for future evolutionary outcomes. I agree, that some particularities of the mechanisms to deal with oxidative stress possibly evolved independently in the three predatory protists investigated, but your proposal that they evolved as response to the "photo-toxicity of [potential] photosynthetic prey" is untestable.

Reviewer #1:

1-1. The study presented examines the effects of photosynthetic prey on heterotrophic amoeba. The authors show that photosynthetic activity of the prey affects gene expression profiles of evolutionary distantly related amoeba species in a similar way. These transcriptomic changes are further accompanied by a reduction in phagocytotic uptake and an increased rate of digestion of engulfed prey, suggesting an active reduction of phototrophs inside the predator cells to reduce oxidative stress caused by the prey. These effects are supposedly similar to observations made in facultative symbiotic associations. The authors therefore conclude that feeding on autotrophic prey requires evolutionary adaptations of the predator to cope with prey induced oxidative stress, which might be conducive to the evolution of symbiotic relationships with autotrophic symbionts.

The manuscript is well written and easy to understand. I generally agree with the underlying hypothesis and find it quite compelling. However, I do have strong reservations regarding the experiments and analyses presented and the extent to which they provide conclusive evidence for the findings.

Thank you very much for your evaluation and we have made following revisions according to the comments.

1-2. The transcriptome analysis presented, for instance, is way below the standard I would expect for a RNA-seq analysis. The current manuscript is not clear on how the gene expression analyses were conducted since important information is missing, but it is clear that the level of replication is too low (only 2 replicates per treatment). Further, I find that the differential gene expression analysis has not been performed using standard normalization and statistical analysis methods, but rather that differentially expressed genes were determined on the level of FPKM values and fold change instead. Furthermore, no additional validation of the contigs was performed using qPCR. With regard to RNA-seq analysis there're several other problems. I'd recommend performing the experiment again using at least 3 biological replicates per treatment and any of the standard analysis pipelines using raw counts for instance (e.g. Trinity, edgeR, kallisto,

etc.). Interestingly, it appears that the authors actually used edgeR to determine enriched GO categories.

Furthermore I'd expect a qPCR validation of the genes identified to respond to the treatment (e.g. myosin, Glutathione peroxidase, etc.) as independent validation of the RNA-seq results as mentioned above.

Thank you very much for the important comment. For *Naegleria*, we performed additional series of cultures on respective conditions and RNA-seq analyses. As a result, we obtained three biological replicates (cultured independently on different days with the predator cells newly prepared from the original frozen stock and bacterial prey prepared at the time of use) (Supplementary Fig. 3).

As described in the previous manuscript, *Acanthamoeba* sp. and *Vannella* sp. growth became unstable after long-term storage, we could not get additional data except for another replicate of *Acanthamoeba* sp. cultured with green prey (the RNA was extracted previously and stored).

In summary we could prepare data of three biological replicates for *Naegleria* sp. in all culture conditions and *Acanthamoeba* sp. with green prey (Supplementary Fig. 3). As suggested, the upregulated and downregulated contigs were extracted by edgeR using the raw count data with the criterion of FDR < 0.01 based on the three independent biological replicates (Figs. 3 and 4; supplementary Tables 3-6).

We also validated the RNA-seq results by quantitative RT-PCR in *Naegleria* sp. (the reason for lack of *Acanthamoeba* and *Vannella* are explained above). The quantitative RT-PCR validation was performed for the genes discussed in the text (Fig. 4 and Supplementary Fig. 6).

1-3. The material and methods part should further provide information on the number of mapped reads per replicate and more detailed information on the way the experiment was actually performed. Were experiments performed on different batches to control for

potential batch effects? At what time points were the different samples taken? This is important to assess if similarities in expression profiles might stem from sampling time point biases i.e. samples taken at the same time point might show higher similarity simply because they were taken at the same time.

Thank you for the important comment. We have summarized the number of mapped reads per sample in Supplementary Table 2. Although the number of reads per replicate varied because the culture and RNA extraction of respective replicates were performed on different days, the difference in number of mapped reads in each comparative pair (i.e., light vs. dark; w/o vs. w/ H₂O₂) was less than twofold except for RNA-seq data of *Vannella* sp. with pale *S. elongatus* prey (light/dark = 2.58) (Supplementary Table 2) (Pages 34-35, lines 718-725).

We have summarized the timing of preparation of respective samples of different culture conditions and replicates in Supplementary Fig. 3 and described the summary in the material and methods part (pages 32-33, lines 672-690). Independent replicates for each condition were obtained from samples that were cultured on different days from one another. In each replicate, samples to be compared were prepared from cultures at the same time (i.e., numbers 1, 2, and 3 of each predator species in Supplementary Fig. 3) (pages 32-33, lines 672-690).

1-4. Similarly, I cannot find important information on the other experiments. How many replicates were used per treatment in the different feeding experiments? The manuscript provides this information for some experiments (e.g. Fig. 2 A) but not all.

We have added information about replicates (legends for Figs. 1e, 2a, 3, 4, 5, and 6b and Supplementary Figs. 1, 2, 4, 5, and 6).

1-6. As for the results presented, the authors state that expression profiles are similar between the different amoeba strains; however, I cannot find an analysis showing this (e.g. PCA, overlap of diff. expressed genes, etc.). Instead the author present

differentially expressed genes of selected GO categories and a Venn diagram without corresponding numbers for the overlap. Based on the list of enriched GO terms listed in Supp. Table 1-6 looks like GO terms might have been “cherry picked” i.e. it seems a bit as if the authors picked a handful of terms out of a quite extensive list . Looking at the supplementary data providing the full list of GO terms (Supp. Table 1-6) further substantiates this impression.

Thank you for your pointing it out. As pointed out, “profiles are similar among three species” in the previous manuscript was an overstatement. We have revised the text as follows. “Upon illumination when feeding on photosynthetic prey, there was some transcriptomic up- or downregulation in common among these evolutionarily distant predators at the level of functional classification and, in some cases, at the gene level” (page 13, lines 261-264).

We have added the KEGG classification of up- and down-regulated contigs (FDR < 0.01; edgeR; three biological replicates) in *Naegleria* sp. and *Acanthamoeba* sp (Fig. 3b). Based on the statistical analysis (FDR < 0.01), number of the GO terms enriched in up- and down-regulated contigs reduced compared with the previous manuscript (Fig. 3a and Supplementary Tables 3-6). As explained in our reply to the comment 1-2, the number of replicates for *Vannella* sp. was only two. Therefore, we could not apply any statistical tests to *Vannella* sp. data. Therefore, we have omitted the *Vannella* sp. data from Figs. 3 and 4 but the data were shown in Supplementary Fig. 6b and Supplementary Fig. 4 for reference.

1-7. With regard to the effects of ROS, the authors have used H₂O₂ as comparison, for instance, but no experiment has been performed to show that ROS levels are indeed elevated when photosynthetic prey is ingested and that ROS levels are reduced by the, supposedly, increased digestion rate. Several dyes exist to perform in vivo staining and measurement of ROS levels (e.g. CellROX) and I suggest including such an analysis.

Thank you for your important suggestion. Actually, before submission of the previous manuscript, we attempted to detect ROS generation by prey inside *Naegleria* sp. cells

upon illumination with CellROX Green and Singlet Oxygen Sensor Green (Thermo Fisher Scientific). However, because of the strong fluorescence by pale and green prey stained with CellROX regardless of presence or absence of illumination or the fluorescence of Oxygen Sensor Green that increased with light alone without *Naegleria* sp. cells and prey, we could not reliably evaluate ROS generation in the cells.

We also compared the H₂O₂ concentration in the medium (with the Amplex Red Enzyme Assay; Thermo Fisher Scientific) between co-culture of *Naegleria* sp. with green prey in the dark and that in the light, but we did not detect any significant difference.

In the previous manuscript these details were not described but it was simply stated “because of these limitations, we could not reliably quantify ROS generation inside amoeba cells despite several attempts.” We have modified this part (page 22, lines 452-462). In addition, we have toned-down the statements that the phototoxicity is due to ROS generation

Although we have not been able to show ROS generation by photosynthetic prey inside predator cells, we have shown (i) cell death upon exposure to high light (Fig. 2), (ii) upregulation of genes related to oxidation/reduction and DNA repair in the excavate and amoebozoans feeding on green prey upon illumination (Fig. 3, Supplementary Fig. 4 and Supplementary Table 3), and (iii) the result that the majority of transcriptome change was shared with responses to ROS treatments (Fig. 4a) suggest that photosynthetic prey exhibits phototoxicity to unicellular colorless predators, which is most likely attributable to oxidative stresses (page 22, lines 463-468).

1-8. Finally, I'd like to comment on the comparisons to endosymbiotic associations. It is not clear if ROS is indeed the trigger for symbiont expulsion or degradation or if ROS is rather an indicator of dysfunctional metabolic interactions i.e. photoinhibition leading to reduced carbon transfer rates and dysfunctional metabolic regulation of the symbiotic relationship. While I agree that the phenotype (reduction of endosymbionts in the tissue) is similar, it cannot be excluded that the underlying cause and mechanism is different. I

think it would take additional experiments to show that these processes are indeed comparable.

Thank you for your important comment. We agree that. We have revised the statements as suggested (page 24, lines 497-510). In addition, we have added the discussion about a possibility that an enzyme for chlorophyll degradation/detoxification, which is shared by photosynthetic organisms, was acquired in prey-predator relationships through horizontal gene transfer (page 24, lines 511-526).

1-9. In summary, while I do find the hypothesis presented interesting and compelling, I do not feel that the results presented provide sufficiently strong evidence given the severe experimental and analytical flaws.

Thank you very much for your important comments. We have revised the MS according to the comments as above.

Reviewer #2:

2-1. This study aims to prove that when a predator is fed with photosynthetic prey under illumination, the photosynthetic oxidative stress of the prey is effecting the metabolic activity of the predator. This consideration could improve our understandings of the evolutionary establishment of photosynthetic eukaryotes.

The finding suggested in this study are, in my view, novel. Reading this study has been very fascinating. The Figure 5 proposes a very interesting model. If published this study will attract much interest. Before publication, in my view, the authors should be able to prove the following points.

Thank you very much for your evaluation and we have made following revisions according to the comments.

2-2.

i)predator without prey does not grow (R0)

ii) predator + dead* photosynthetic prey, in the dark can grow at a certain rate (R1)

iii) predator + alive photosynthetic prey, in the dark can grow at a certain rate (R2)

With R1 and R2 should be similar

iv) predator + dead* photosynthetic prey, in the light (low**) can grow at a certain rate (R3)

v) predator + alive photosynthetic prey, in the light (low**) can grow at a certain rate (R4)

With R4 should be > R3

With R3 should be similar to R2 and R1

vi) predator + dead* photosynthetic prey, in the light (high**) can grow at a certain rate (R5)

vii) predator + alive photosynthetic prey, in the light (high**) can grow at a certain rate (R6)

With R4 should be > R6

With R5 should be similar to R3, R2 and R1

*non photosynthetically active

**low light should be able to power the photosynthesis of prey without generation of oxidative stress. High light should be able to power the photosynthesis of prey but with generation of oxidative stress

Figure 2A, to better compare those three conditions (dark, low light and high light) please make the Y axis identical.

Please provide the relative negative control (therefore *Naegleria* without feeding).

Also, please provide three equal graph showing the cell growth of *S. elongatus* without the *Naegleria*.

Thank you for your suggestions. We have revised Fig. 2a in which Y-axis is identical for all conditions (Fig. 2a and supplementary Figs. 1 and 2). In addition, we have added the results of the cell growth of *S. elongatus* without the *Naegleria* sp. (Supplementary Fig. 1). In the time range of the measurements (0–360 min; Fig. 2a), the increase of the number of green *S. elongatus* cells in the low-light ($\times 1.10 \pm 0.02$) and high-light ($\times 1.11 \pm 0.02$) conditions was relatively slow and did not differ markedly from that in the dark ($\times 0.99 \pm 0.04$) in the inorganic medium with limited concentrations of nutrients (Supplementary Fig. 2). Likewise, the number of pale *S. elongatus* changed little in the

dark ($\times 1.00 \pm 0.03$), in the low-light ($\times 1.02 \pm 0.06$), and high-light ($\times 0.99 \pm 0.04$) conditions (Supplementary Fig. 2). Thus, in this assay, the effect of *S. elongatus* growth and its difference depending on the culture conditions and between the green and pale cells on *Naegleria* sp. growth could be ignored. We have added the description to the text (page 9, lines 165-174).

Because *Naegleria* sp. (also *Acanthamoeba* sp. and *Vannella* sp.) is a heterotrophic organism, it did not grow in inorganic medium without bacterial prey, as well known, and gradually formed cysts after the removal of prey as previously described. We have added this information to the text (pages 8-9, lines 163-165).

Regarding the “dead photosynthetic prey”, dead *S. elongatus* cells still contain photosynthetic pigments (chlorophyll *a* and phycobilins) although the cell probably does not photosynthesize when proteins in the photosynthetic apparatus are denaturalized. However, these photosynthetic pigments are probably still phototoxic to predators. Therefore, in this study, we have applied “pale” *S. elongatus* cells, which almost lack photosynthetic apparatus, including chlorophylls and phycobilins, and photosynthetic activity, rather than dead *S. elongatus* (pages 7-8, lines 137-143).

2-3. Could the authors provide some data about the amount of oxygen in the three conditions (dark, low light and high light)?

Thank you for your suggestion. In the coculture of unicellular predators and bacterial prey, we left Petri dishes to stand without agitation or aeration (Fig. 1e). This was important for bacterial prey to stay on the bottom of the dishes and for predators to easily ingest bacterial prey. Therefore, the dissolved oxygen concentration was probably heterogeneous in the liquid medium and varied depending on the microenvironment. Therefore, we could not examine the oxygen concentration in the coculture.

2-4. I am not sure about the use of *E. coli* cells incubated in acetone with (Chl) or without (w/o Chl) chlorophyll *a*, dried, and then rehydrated. I do not expect the Chl

transferred inside the *E. coli* able to do any photochemistry. Autofluorescence is OK, but not photochemistry.

Thank you for the important comment. We asked a colleague who is a specialist of photochemistry of chlorophylls and their derivatives. As in Kashiwama and Tamiaki (2014) (newly cited), the emission of fluorescence from *E. coli* stained with chlorophyll *a* (Fig. 1f) indicates that at least some of the chlorophyll *a* molecules on the *E. coli* cells were transformed to an S₁ state by excitation light, which can transform to T₁ and produce ¹O₂/O₂⁻ by releasing energy/electrons to an oxygen molecule, leading to relaxation to S₀²⁴. In addition, it is possible that chlorophyll *a*, which is released from *E. coli* in predator cells during digestion, and its derivatives also became toxic to predator cells. We have added these explanations to the text (page 14, lines 289-294).

2-5. Figure 4. Could the author explain why the decrease in FM1-43 fluorescence should be a proxy for digestion rate? And why this rate should be related with the presence of light.

FM1-43, which is water-soluble and nontoxic to cells, is not fluorescent in aqueous phase but becomes fluorescent upon its insertion into the cell membrane. The fluorescence of FM1-43-stained green and pale prey was stable for at least a week in our experimental conditions. Thus, the reduction of fluorescence after the prey was ingested by *Naegleria* sp. cells probably reflects digestion of the cell membrane of the prey by *Naegleria* sp. In our experimental conditions, the fluorescent intensity of the free prey outside *Naegleria* sp. cells was kept almost constant both in the dark and light in the time range of the measurements (Fig. 5d). Thus, the reduction of fluorescence was attributable to digestion of cell membrane rather than bleaching of the FM1-43 fluorescence in the cell membrane of prey. We have added the explanation to the text (page 18, lines 363-368 and 373-377).

Reviewer #3:

3-1. The authors present an exciting study exploring the physiological response of amoeba when feeding on photosynthetic prokaryotes. The authors explored this question in a very thorough and careful way, testing various hypothesis and subsidiary questions. The authors developed clever experiments such as the culture with fluorescent beads to practically quantify food uptake. This helps to produce a manuscript not only based on transcriptomic analysis but on empirical data as well. This work opens interesting perspectives on the evolution of photosynthesis in eukaryotes through endosymbiosis.

The English and writing is in general good. However it is sometimes hard to follow the logic of the authors as they present results in a little confused way. Notably, I suggest that the discussion on the transcriptomic analysis should be entirely rewritten for a better understanding.

Thank you very much for your evaluation and we have made following revisions according to the comments. Regarding the discussion on the transcriptomic analyses, please see our reply to the comment #3-4.

3-2. Page 6 line 104-119: The authors studied the growth rate of amoeba under different culture conditions. However the authors did not actually calculated a growth rate. The authors should present a statistical analysis of their results. This might change the discussion. For example, based on the histograms presented on the Figure 2, I am not convinced that the growth rates between the “Dark” and “Low light” are similar each other. It seems that even in “Low light” the growth rate is stagnant for the first 90 minutes. Statistical comparison of the different rates could help to more clearly understand the reaction of amoeba to the culture conditions. The author should also present all the histograms using the same y-axis in order to ease the visual comparison between the different cultures.

Thank you for your helpful suggestion, we have calculated the growth rate of *Naegleria* sp. and performed statistical comparison (Fig. 2a; the raw data has been moved to

supplementary Fig. 2). The Y-axis was unified as suggested. We have modified the description and interpretation of the results in the text (page 9, lines 175-184).

3-3. Page 7 line 134-137: This sentence is confusing and could be rewritten. The authors are explaining how they searched for “false positive” transcripts identified from their experiment.

We have deleted the sentence because of the reconstruction of the transcriptomic analysis part. We have newly described the way of comparisons of results obtained from cultures of several different conditions (pages 13-14, lines 269-285).

3-4. Page 7-9 line 121-175: The transcriptome analysis the authors present is impressive for the control experiments that the authors conducted to explore further the expression regulation occurring during feeding of their amoeba isolates. However the different analysis and controls conducted by the authors are presented in a somewhat disordered way and it is hard to understand what set of the transcripts the authors are discussing. I would suggest that the authors more clearly separate the analysis in two different parts. First, the main analysis of the up/down regulated transcripts and exploring the main GO terms represented and species specific regulations. And in a second part, discuss those results in the light of the different control experiments. I think this would help for readers of the manuscript without altering the main results of this transcriptomic analysis (i.e. expression regulation is altered while feeding on photosynthetic prey but most of those regulation can be triggered by oxidative-stress and/or illumination). Rewriting of this part of the manuscript should also be accompanied by a revision of the Figure 3B. The Venn diagrams are not self-explanatory in my opinion and the numbers presented do not add up with the Up/Down regulated transcripts from the main experiment (i.e. feeding on photosynthetic prey under low light).

Thank you for your suggestion. As suggested we have revised the text. First, we discussed the main analysis of the up/down regulated transcripts and classification of GO terms and KEGG functional categories (Fig. 3; pages 10-13, lines 196-264). Then,

we discussed the results based on the different control experiments and responses of specific genes (Fig. 4 and supplementary Figs. 4-6; pages 13-16, lines 269-338). As suggested, we have revised the Venn diagrams (Fig. 4a).

3-5. Finally, a more thorough discussion on the evolution of those regulations would be a nice addition to this manuscript. The authors could for example explore differentially regulated transcripts without annotation that are shared among the three species. Despite not having an annotation exploring the domain encoded and those transcripts could bring interesting evolutionary insights and suggest target for further studies. But this might not be statistically meaningful (see below).

Thank you for your suggestion. The genes that were commonly up- or downregulated in the three excavate and amoebozoan species (myosin etc.) were already described in the previous manuscript. As suggested, we further searched commonly changed genes. As a result, we found genes encoding proteins that possess both Rieske [2Fe-2S] iron-sulfur domain and pheophorbide *a* oxygenase (PAO)-like domain were upregulated upon illumination with feeding on green prey in the three excavate and amoebozoan species. We have added the results (Fig. 6) and discussion (pages 18-21, lines 382-443) to the manuscript. These proteins of horizontal gene transfer origin are likely involved in detoxification of chlorophylls or their derivatives (pages 24-25, lines 507-519).

Regarding the number of replicates and statistical analyses, please see our reply (revisions) to the comments 1-2 and 1-3.

3-6. Page 21 line 418-421: The entire transcriptomic analysis are based on duplicate and not triplicate. It is generally accepted that triplication is the minimum for statistically reliable transcriptomic analysis (although triplicates are actually still not sufficient, see <https://www.ncbi.nlm.nih.gov/pmc/articles/PMC4878611/> for a discussion on this). Therefore I am not sure that the transcriptomic analysis, despite being well designed and interesting (see above), are statistically meaningful. However this does not constitute a reason to reject the paper in my opinion as the most interesting results

(again in my opinion) are based on empirical observation (e.g. feeding rate) and globally are in accordance with the main trends observed in the transcriptomes.

Please see our reply (revisions) to the comment #1-2 and #1-3.

Minor comments:

3-7. Page 3 line 32: References concerning the dates advanced by the authors should be added.

As suggested, we have added a reference (page 3, line 35).

3-8. Page 3 line 48: The term “transparent” might be not suited. I suggest the authors use the term colorless when referring to non-photosynthetic organisms.

As suggested, we have changed “transparent” to “colorless” (pages 5 and 22, lines 85 and 467).

3-9. Page 4 line 63: “suggesting that the responses” could be changed “suggesting that these responses”.

3-10. Page 4 line 69: “which is observed”.

Thank you for these comments. After substantial revisions, we have checked the grammar throughout the text.

3-11. Page 5 line 82: The authors used a blastn search to identify the amoeba species they isolated. The best hits scores should be presented in a supplementary table. Furthermore the authors could present a 18S rRNA phylogenetic tree including their isolates to determine more clearly their identification.

As suggested, we have added a table showing the best hits of BLASTn searches of 18S rDNA sequences of respective amoebae strains (Supplementary Table 1). We think that the phylogenetic tree would be redundant for this manuscript. So, we did not add the phylogenetic analysis.

3-12. Page 5 line 87 Page 7 line 121 Page 13 line 255: “evolutionarily distantly related” could be replaced simply by “distantly related” or “evolutionarily distant”.

As suggested we have changed them throughout the text.

3-13. Page 8 line 154-155: “but not in unstained E. coli upon illumination”.

Thank you for the comment. After substantial revisions, we have checked the grammar throughout the text.

3-14. Page 10 line 186-191: I think cox15 is also a component of the cytochrome c oxidase assembly (at least in yeast) and its up-regulation could have an effect on mitochondrial respiration as well. The authors should explore on this.

Thank you for your suggestion. We have added the information to the text (page 15, lines 310-312).

Reviewer #4:

4-1. The report presents an interesting study about the cellular and molecular responses of three different types of predatory protists after engulfing cyanobacterial preys. The main hypothesis is that the protist cellular systems that cope with oxidative stress have an important role contending with the production or reactive oxygen species (ROS) by the photosynthetic cyanobacterial preys. The working hypothesis has evolutionary implications, because a widely accepted scenario posits that the origin of primary

photosynthetic organelles (i.e., plastids) likely involved similar ecological interactions between predatory protists and cyanobacterial preys. It has been discussed largely the importance of the prey survival mechanisms in the establishment of long term endosymbiotic associations, but this contribution of Uzuka and collaborators focuses on the other side of the coin: how do the unicellular host copes with the putative toxicity, given the ROS release, of the photosynthetic partner?

Uzuka and collaborators designed a series of experiments to evaluate at different levels the responses of predatory protists to different intensities of photosynthetic activity in their cyanobacterial preys. The authors suggest that in the different studied protists similar gene expression responses occur to regulate the phagotrophic activity, prey digestion and mechanisms of protection against oxidative stress. It is suggested also that these gene expression changes are the predator responses to limit the potential damage caused by the cyanobacteria-produced ROS, to reduce the capture of more cyanobacteria and to simultaneously accelerate the digestion of the captured photosynthetic cells.

Overall the study presents experimental results that have important implications to better understand cellular mechanisms involved on early stages of endosymbiotic associations with photosynthetic partners. However, I have a series of concerns that in my view have to be addressed before consideration for publication in Nature Communications.

Thank you very much for your evaluation and we have made following revisions according to the comments.

Major concerns

4-2. The Introduction section should be re-written using more precise ideas and rigorous organization. The second paragraph uses loose sentences to describe an important piece of the context: the general hypothesis about the origin of primary plastids. Please expand this point and describe the different ideas detailed in plenty of recent publications. For example, the opening sentence “It is believed that chloroplasts were

established through predation and the temporary retention of photosynthetic prey” is an oversimplification of the sophisticated scenarios that have been proposed, which are relevant to justify both the tested hypothesis and the design of the experimental work. Then, the use of the “herbivorous” adjective is odd for protists that actually are referred in the text as consumers of cyanobacteria, which are not plants or herbs. The third paragraph includes mostly description of results (lines 58-66) and sentences that seem more adequate for discussion (Lines 67-72). Please revise the entire introduction section.

Thank you for your important suggestion. We have re-written the introduction according to your suggestion (pages 3-6, lines 42-103). We agree that “herbivorous” is not suitable and have deleted “herbivorous”.

4-3. Do the three studied protist species regularly consume *Synechococcus elongatus* in their natural habitat? It is mentioned, and illustrated, that the sampling site is dominated by ‘microalgae’ (photosynthetic eukaryotes). Is there any specific information about the presence of *Synechococcus elongatus* in the same water body? Why did you use cyanobacteria instead of the apparently abundant algal preys? Clarify and justify your decision.

Thank you for your pointing it out. In the natural habitat where we isolated the three excavate and amoebozoan species, we observed *Synechococcus*-like algae, but they were rare in the water samples. First, we attempted to isolate and culture a non-ciliated green alga that dominated there (Fig. 1b; it did not swim and was suitable for predators to feed on), but we failed to culture it. Thus, we used the cyanobacterium *S. elongatus*, in which the genome sequence was completely sequenced and the procedure to prepare pale cells had been developed. The genome sequence information was important for this study because we had to remove the contaminated prey sequence from RNA-seq reads to extract sequences of the excavate and amoebozoans. The preparation of pale cells was important to examine the effect of the photosynthetic trait and photosynthetic pigments of prey on the unicellular predators by comparing co-culture with normal

blue–green and pale *S. elongatus* prey. We have added these explanations to the text (pages 7-8, lines 121-143).

Therefore, the co-culture systems that we used do not completely mimic conditions in the wild. The changes in the mRNA levels of some genes in predators observed here are likely specific to *S. elongatus* and likely do not occur in the predators feeding on other microalgae in natural habitats. However, as discussed above, the majority of the transcriptomic changes upon illumination in *Naegleria* sp. feeding on green *S. elongatus* are attributed to ROS and/or ingestion of chlorophyll *a*, which should be common to photosynthetic prey containing chlorophyll *a* and photosynthetic apparatus, regardless of the lineage and species. We have added this point to the discussion (pages 22-23, lines 469-479).

4-4. How did you establish the time of exposure to bacterial preys in the co-cultivation experiments (Figure 2)? Please explain.

As shown in Fig. 2b, we observed round dying cells only in the culture with the green prey under high-light conditions 60 min but not 180 min after the onset of light exposure. Based on this observation, we examined the number of *Naegleria* sp. cells 90, 180, and 360 min after the onset of light exposure calculated the growth rate (Fig. 2a). We have added this explanation to the text (page 8, lines 158-163).

Because the results of *Naegleria* sp. growth (Fig. 2) suggested that the cells had acclimated to the phototoxicity of green prey by 90 min after the onset of illumination (page 9, lines 177-184). Therefore, we compared transcriptome 60 min after the onset of illumination with that immediately before the illumination. We have added the explanation to the text (page 10, lines 202-205).

4-5. It is likely that the captured cyanobacteria are photosynthetically active under light conditions, but it will be important to evaluate the actual carbon fixation rate directly because that aspect is another important part of the general hypothesis of plastid origins

by endosymbiosis: the prey likely releases fixed carbon compounds into the host/predator cytoplasm and that also must play a key role in the digestion response of the host. Is there any active or passive transport of carbon compounds from the cyanobacteria to the protist cell? Is this different in “green” and “pale” prey?

Thank you for your pointing it out. As pointed out, it is known that microalgae including cyanobacteria excrete organic compounds under certain conditions. A certain carbohydrates are likely excreted by *S. elongatus* prey in amoeba cells after ingestion for a short period before being digested. However, *S. elongatus* begun being digested immediately after ingestion by predator cells (e.g. Fig. 5). Even if a certain organic compounds, which are excreted by *S. elongatus* prey in phagosomes of predators for a short period, is actively absorbed by amoebae, the effect as nutrients for predators would be much less than the total organic compounds obtained by digesting the *S. elongatus* whole cell.

Thanks to your important comments, we noticed that it was worth comparing difference in transcriptome between predators with green prey and those with pale prey. However, we could not find any candidates for carbohydrate transporters that are expressed at higher level in *Naegleria* with green prey than that with pale prey.

Besides the study on prey-predator relationships, our laboratory is also studying kleptoplasty, in which a non-photosynthetic dinoflagellate ingests a microalga and retains and utilizes its chloroplast up to 30 days before digesting the ingested chloroplast. In this case, we have found a sugar transporter encoded by the dinoflagellate genome that is upregulated upon ingestion of the alga and illumination. Therefore, we think that active transport of organic compounds secreted by a prey/endosymbiont would be required for organisms that retain ingested algae for a certain period rather than organisms that start to digest algae immediately after ingestion.

4-6. Did you try to expose protist to “green” cyanobacterial preys under light conditions in the presence of inhibitors of the photosynthesis or compounds that limit ROS

formation by the photosynthetic electron chain? This would be a good experimental control to untangle the set of protist genes expressed exclusively in response the ROS production by the cyanobacterial photosynthetic activity.

Thank you for your pointing it out. We first tried to compare the behavior and transcriptome of predators with green prey w/ or w/o DCMU (a widely used inhibitor of photosynthetic electron flow) to examine the effect of photosynthesis of prey on the predators. However, we found that DCMU was toxic to amoebae we used. Predator cells started to die by the addition of DCMU regardless of presence or absence of cyanobacterial prey both in the light or dark. Therefore, we changed the strategy to the comparison of amoebae between the coculture with green prey and that with the pale prey. Pale prey, prepared by prolonged nitrogen starvation, exhibits degradation of the bulk of its photosynthetic apparatus, including chlorophylls and phycobilins (Fig. 1d, e) and has lost almost all of its photosynthetic activity (~0.1% compared with that of normal green cells) (Sauer et al., 2001). We have described these points to explain why pale prey was prepared and used in this study (pages 7-8, lines 131-143).

4-7. I do not understand clearly why ‘pale’ cyanobacterial preys were prepared. It is formally possible that the actual endosymbiosis that gave rise to primary plastids occurred with ‘pale’ endosymbionts under low light if the nutrient flux between both partners allowed a stable endosymbiotic association. Under such scenario, the level of ROS production by the cyanobacteria would not have the same impact. Is that the reason why you decided to include ‘pale’ cyanobacterial preys in your experiments? Please explain the decision in more detail.

Please see our reply to the comment 4-6.

4-8. Figures 2 and 3 are unnecessary busy. In my opinion the number of panels is excessive. Is this because a limitation in the number of allowed figures? The figures are hard to follow and I strongly recommend more concise streamlined versions. For example, panels A, B, C and G of Figure 2 are in my opinion superfluous.

We have revised the contents and construction of Fig. 3 and divided the contents to two figures (Figs 3 and 4).

Minor comments

4-9. Use 'plastids' (a generic term) instead of 'chloroplasts' (i.e., the plastids of green algae and plants).

As suggested, we have changed “chloroplasts” to “plastids” throughout the text.

4-10. 'Amoeba' is a loose term with no phylogenetic significance. I would suggest you use 'excavate' or 'amoebozoan', depending on the species, instead of 'amoeba'.

As suggested, we have changed “amoebae” to specific taxonomic names where appropriate.

4-11. Line 87- “evolutionally distantly related’ is a vague sentence. Please be more specific.

We have changed “evolutionally distantly related’ to “distantly related” or “evolutionarily distant”.

4-12. Line 112- Why is the description of the cells shape relevant? Lines 115-116. This is an interesting but speculative sentence. You need to provide evidence of that.

We have revised the text and added the explanation (pages 9-10, lines 185-194).

13. Lines 235-236. “This difference is probably because the pale prey, which was prepared by nitrogen starvation, was more easily digested by amoebae”. What is the

evidence to suggest that? Overall, the text contains several speculative sentences that have to be reduced to a minimum, or eliminated.

Thank you for your pointing it out. We agree that and have removed the description.

14. Lines 260-263 – This is a deterministic statement. All free-living cells, predatory or not, have mechanisms to cope with oxidative stress, but those are not ‘prerequisites’ for future evolutionary outcomes. I agree, that some particularities of the mechanisms to deal with oxidative stress possibly evolved independently in the three predatory protists investigated, but your proposal that they evolved as response to the “photo-toxicity of [potential] photosynthetic prey” is untestable.

Thank you for your pointing it out. We agree your comment. We have revised the statement (page 23, lines488-490).

Finally, we have added Dr. Yusuke Kobayashi and Dr. Ryo Onuma as co-first authors of this manuscript. Because Dr. Akihiro Uzuka left our lab. after getting his Ph.D., Dr. Kobayashi has taken over the experiments and Dr. Onuma has taken over the analyses of RNA-seq data for the revision.

Reviewers' Comments:

Reviewer #1:

Remarks to the Author:

The authors have addressed most of my comments and added additional data and validations as suggested. While not everything has been or could be addressed, I commend the authors for the additional work they invested into their study. However, I still do have a few comments and minor changes.

First off, the authors now included additional biological replicates for their RNAseq and used an appropriate analysis pipeline. They also validated expression changes of selected genes via RT-PCR, but I am missing some explanation on how the results of these validations compare to the expression changes observed for these genes in their RNAseq analysis. Furthermore, looking at the RT-PCR in Supp. Figure 6, it appears that expression for these genes responds more to light rather than the type of prey or the presence of chlorophyll. None of these things appear to be discussed appropriately, although similar observations for the RNAseq results are mentioned in the result section. Further, I was hoping for a better comparison of the expression profiles e.g. a PCA as I previously suggested, but instead the authors still provide a Venn diagram instead. For the interspecies comparison the authors rely on higher level GO terms from the enrichment analysis, but it would be nice to have a broader overview of similarities on a gene level other than the few genes discussed in the results section.

Regarding the detection of ROS using a fluorescent probe (e.g. CellROX) the authors report that this has been tried but that autofluorescence did not allow the quantification. I find this finding quite odd since I have used CellROX on algae as well as naturally fluorescent organisms and I did not observe such problems. Furthermore, CellROX can be obtained both as green as well as orange fluorophore to mitigate problems with autofluorescence. It would have been nice to add this independent validation.

Line 84 "photosynthesis" or "phototroph

y" or "the ability to photosynthesize" instead of "phototropic ability"

Line 126 complicated sentence structure, maybe "for which a genome assembly is available"

Lines 240-246 sentence is long and complicated, consider breaking it into multiple sentences

Line 288 green light for excitation of chlorophyll or rather blue light?

Line 295 -304 long and complicated sentence, consider breaking it into several simpler sentences

Line 432 I don't think "were suggested" is the correct or best way to phrase this. Maybe "Phylogenetic analysis suggests that 2 genes ... have been acquired .."

Reviewer #2:

Remarks to the Author:

The improvement and changes implemented in the manuscript of Uzuka et al., are, in my view, sufficient for recommending publication on Nature Communication.

Reviewer #3:

Remarks to the Author:

The manuscript presented was greatly improved from its original version. The authors addressed my main concern on the manuscript that was the transcriptomic analysis part of the manuscript. The authors clarified the presentation of their results and at least for the experiment with green prey obtained triplicate. The authors also toned down some of their statements, especially in regard to the lack of replication and/or validation of the transcriptomic by qPCR in the case of *Acanthamoeba*. Finally the author added an interesting paragraph on the PAO-like transcripts that were upregulated in all the three species considered while feeding on green prey under light. This analysis was suggested so that the authors could broaden their discussion, notably on the evolutionary implications of their experiments.

Threshold on bootstrap support are really partial and will change between people. In my opinion, a value of 78 is not convincing and especially in this kind of analysis, I would not draw conclusions on nodes with support lesser than 90. Furthermore, I think that the important message of this tree was that the origin of the PAO-like in *Naegleria* and *Acanthamoeba* was different than the one in the primary and secondary plastid lineages. This opens the evolutionary discussion on the establishment of a plastid in my opinion. However, the authors' discussion opposes the EGT and HGT (page 25 line 520-523). More than that I think these results highlight the dynamic nature of the process and the multiple small steps that took place during the establishment of a primary endosymbiosis.

Minor comments:

Page 11 line 217: The authors indicate the percentage of genes that showed regulation in their transcriptomic study. However, without mention of the number of transcripts assembled this percentage is a little informative. Furthermore, the notation is unclear. Did 8% of the transcripts were upregulated and 3% downregulated in *Naegleria* and 7% upregulated and 2% downregulated in *Acanthamoeba*? I think this could be rewritten in a more clear way although the Figure is clearer and helps to understand

Reviewer #4:

Remarks to the Author:

I am satisfied with the answers and modifications that the authors have made after my questions and comments. However, the abstract, mainly, but also the text body in general, are still including loose and even ambiguous terms, such as "microalgae", "Synechococcus-like algae", and "microheterotrophs".

Cyanobacteria (i.e., *Synechococcus*) are not algae, then strictly they are excluded from the "microalgae" definition. Please use along the entire text the proper taxonomic names to refer the species used in your study instead of, for example, "microalgae". Then, the "microheterotroph" definition can include also heterotrophic bacteria. Please replace those ecological concepts that have no evolutionary or taxonomic relevance. The entire text has to be revised thoroughly to avoid the use of those loose biological definitions.

I have no additional comments.

Reviewer #1:

1-1. The authors have addressed most of my comments and added additional data and validations as suggested. While not everything has been or could be addressed, I commend the authors for the additional work they invested into their study. However, I still do have a few comments and minor changes.

Thank you very much for your evaluation and we have made following revisions, according to the comments.

1-2. First off, the authors now included additional biological replicates for their RNAseq and used an appropriate analysis pipeline. They also validated expression changes of selected genes via RT-PCR, but I am missing some explanation on how the results of these validations compare to the expression changes observed for these genes in their RNAseq analysis.

Thank you for pointing it out. We have summarized how qRT-PCR results support the results of RNA-seq (page 11, lines 224-226; pages 15-16, lines 312-324.).

1-3. Furthermore, looking at the RT-PCR in Supp. Figure 6, it appears that expression for these genes responds more to light rather than the type of prey or the presence of chlorophyll. None of these things appear to be discussed appropriately, although similar observations for the RNAseq results are mentioned in the result section.

Thank you for pointing it out, we have added the discussion on the upregulation of some genes by light stimuli regardless of photosynthetic traits of prey (page 25, lines 522-531.). Please also see our reply to the comments #1-4.

1-4. Further, I was hoping for a better comparison of the expression profiles e.g. a PCA as I previously suggested, but instead the authors still provide a Venn diagram instead.

Thank you for your important suggestion. We performed the t-distributed Stochastic Neighbor Embedding (t-SNE) analysis to compare whole transcriptomic changes in respective conditions in a two dimensional map (Fig. 4b).

The changes upon illumination with reduced (with pale prey) and no (without prey, with unstained *E. coli*) photosynthetic traits of prey were similar (Fig. 4b). In contrast, the transcriptomic change with green prey was placed at a different position (Fig. 4b), suggesting that the transcriptomic change when feeding on green prey is not solely attributable to the light stimulus. In addition, the change with chlorophyll-stained *E. coli* was positioned close to the change with green prey (Fig. 4b). Although the changes with RB or upon H₂O₂ addition largely deviated among three sets of replicates, they were positioned between the change with green prey and those with reduced or no photosynthetic traits of prey (Fig. 4b). We have added the explanation to the text (pages 16-17, lines 341-356.).

1-5 For the interspecies comparison the authors rely on higher level GO terms from the enrichment analysis, but it would be nice to have a broader overview of similarities on a gene level other than the few genes discussed in the results section.

Thank you for your important suggestion. As suggested, we compared the changes of mRNA levels upon illumination for all orthogroups shared by *Naegleria* sp. and *Acanthamoeba* sp. (Fig. 3b). As a result, both the comparison of the whole orthogroups (2,585 groups) and that of only one-to-one orthologs (i.e. one orthogroup was composed of single copy genes rather than multiple homologs of respective species; 1,823 orthologs) exhibited a weak correlation (Spearman rank correlation coefficient; $\rho = 0.24$ and 0.25 , respectively) between the two species (Fig. 3b). The result suggests that a certain degree of transcriptomic change was shared between the two species. We have added these explanations to the text (pages 11-12, lines 227-236.).

1-6. Regarding the detection of ROS using a fluorescent probe (e.g. CellROX) the authors report that this has been tried but that autofluorescence did not allow the

quantification. I find this finding quite odd since I have used CellROX on algae as well as naturally fluorescent organisms and I did not observe such problems. Furthermore, CellROX can be obtained both as green as well as orange fluorophore to mitigate problems with autofluorescence. It would have been nice to add this independent validation.

Thank you for the comment. We again tested CellROX Green and Orange. When CellROX Green or Orange was applied after free green prey that had not been ingested were removed from co-culture of *Naegleria* sp. and the green prey culture, mitochondrial nucleoids or mitochondria of *Naegleria* sp. cells emitted weak fluorescence, respectively, both in the light and dark. However, we could not detect any difference in the fluorescent intensity between the two conditions. We have added the explanation to the text (page 24, lines 510-515.).

1-8.

Line 84 “photosynthesis” or “phototrophy” or “the ability to photosynthesize” instead of “phototropic ability”

Line 126 complicated sentence structure, maybe “for which a genome assembly is available”

Lines 240-246 sentence is long and complicated, consider breaking it into multiple sentences

Line 288 green light for excitation of chlorophyll or rather blue light?

Line 295 -304 long and complicated sentence, consider breaking it into several simpler sentences

Line 432 I don’t think “were suggested” is the correct or best way to phrase this. Maybe “Phylogenetic analysis suggests that 2 genes ... have been acquired ..”

Thank you for checking the text carefully, we have modified/fixed the text as suggested.

Reviewer #2:

2-1. The improvement and changes implemented in the manuscript of Uzuka et al., are, in my view, sufficient for recommending publication on Nature Communication.

Thank you very much for your time and evaluation.

Reviewer #3:

3-1. The manuscript presented was greatly improved from its original version. The authors addressed my main concern on the manuscript that was the transcriptomic analysis part of the manuscript. The authors clarified the presentation of their results and at least for the experiment with green prey obtained triplicate. The authors also toned down some of their statements, especially in regard to the lack of replication and/or validation of the transcriptomic by qPCR in the case of Acanthameoba. Finally the author added an interesting paragraph on the PAO-like transcripts that were upregulated in all the three species considered while feeding on green prey under light. This analysis was suggested so that the authors could broaden their discussion, notably on the evolutionary implications of their experiments.

Thank you very much for your evaluation.

3-2. Threshold on bootstrap support are really partial and will change between people. In my opinion, a value of 78 is not convincing and especially in this kind of analysis, I would not draw conclusions on nodes with support lesser than 90. Furthermore, I think that the important message of this tree was that the origin of the PAO-like in Naegleria and Acanthamoeba was different than the one in the primary and secondary plastid lineages. This opens the evolutionary discussion on the establishment of a plastid in my opinion. However, the authors' discussion opposes the EGT and HGT (page 25, lines 520-523). More than that I think these results highlight the dynamic nature of the process and the multiple small steps that took place during the establishment of a primary endosymbiosis.

Thank you for your useful suggestions. According to your suggestions, we have revised the text (page 23, lines 479-487.) and added the discussion (page 28, lines 586-594.).

3-3. Minor comments: Page 11 line 217: The authors indicate the percentage of genes that showed regulation in their transcriptomic study. However, without mention of the number of transcripts assembled this percentage is a little informative. Furthermore, the notation is unclear. Did 8% of the transcripts were upregulated and 3% downregulated in Naegleria and 7% upregulated and 2% downregulated in Acanthamoeba? I think this could be rewritten in a more clear way although the Figure is clearer and helps to understand

Thank you for pointing it out. We have realized that the statement was confusing. We have revised the statement (page 11, lines 220-224.).

Reviewer #4:

4-1. I am satisfied with the answers and modifications that the authors have made after my questions and comments. However, the abstract, mainly, but also the text body in general, are still including loose and even ambiguous terms, such as “microalgae”, “Synechococcus-like algae”, and “microheterotrophs”. Cyanobacteria (i.e., Synechococcus) are not algae, then strictly they are excluded from the “microalgae” definition. Please use along the entire text the proper taxonomic names to refer the species used in your study instead of, for example, “microalgae”. Then, the “microheterotroph” definition can include also heterotrophic bacteria. Please replace those ecological concepts that have no evolutionary or taxonomic relevance. The entire text has to be revised thoroughly to avoid the use of those loose biological definitions.

Thank you very much for your time and evaluation. We have revised the usage of terms indicating certain organisms throughout the text as suggested.

We believe that our manuscript has been improved, thanks to the editors and reviewers,

and hope it is now acceptable for publication in *Nature Communications*. We look forward to your decision.

Yours sincerely,

Shin-ya Miyagishima

Reviewers' Comments:

Reviewer #1:

None

Reviewer #3:

Remarks to the Author:

The authors have addressed my comments on the PAO and improved this section of the manuscript. I think the manuscript holds a very interesting story and is now recommendable for publication.